# Boosting the electron beam transmittance of field emission cathode using a self-charging gate

Dongyang Xiao[1,2], Huanhuan Du[1,2], Leimeng Sun [1] ✉, Xiaochen Suo[2], Yurong Wang[2], Yili Zhang[2], Shaolin Zhang[2], Shuangyang Kuang[3], Fangjing Hu [2], Liangcheng Tu[2,4], Daren Yu[5] & Peiyi Song [2] ✉

The gate-type carbon nanotubes cathodes exhibit advantages in long-term stable emission owing to the uniformity of electrical field on the carbon nanotubes, but the gate inevitably reduces the transmittance of electron beam, posing challenges for system stabilities. In this work, we introduce electron beam focusing technique using the self-charging $SiN_x$/Au/Si gate. The potential of $SiN_x$ is measured to be approximately −60 V quickly after the cathode turning on, the negative potential can be maintained as the emission goes on. The charged surface generates rebounding electrostatic forces on the following electrons, significantly focusing the electron beam on the center of gate hole and allowing them to pass through gate with minimal interceptions. An average transmittance of 96.17% is observed during 550 hours prototype test, the transmittance above 95% is recorded for the cathode current from 2.14 μA to 3.25 mA with the current density up to 17.54 mA cm$^{-2}$.

Carbon nanotubes (CNTs) present numerous advantages as field emission cathode (FEC), such as their ultralow turn-on voltage (~0.4 V μm$^{-1}$)[1] resulting from the low work function (~5 eV)[2], exceptional chemical and physical stabilities[3], as well as high electrical and thermal conductivity[4,5] attributed to the C−C covalent bond, seamless hexagonal network architecture and one-dimension carbon nanostructures[6,7]. These unique characteristics render FEC an ideal electron source for various applications, including field emission scanning electron microscopy (FESEM)[8], field emission display (FED)[9], electron beam (E-beam) lithography (EBL)[10], and X-ray tube[11]. In modern space technologies[12], FEC often serves as the cathode for electric propulsion (EP) system, providing electrons to neutralize the positive ions or positively charged droplets to avoid charging of the spacecraft[13]. Missions like space-borne gravitational wave detection and deep-space exploration request the EP system

to work in long-term (up to 5–10 years) without any possibility of replacement[14], requiring a longer lifetime for the cathode as well as the stability of emission (current fluctuation <10% during the whole lifetime for perfect neutralization[15,16]. Studies show that the Spindt-type cathode cannot fulfill the aforementioned requirements due to the electric field non-uniformity on the CNTs[17]. In this case, CNTs under a higher electric field emit earlier, the rest are shield[18]. Continuous emitting of very small amounts of CNTs results in heating and thermal evaporating of carbon, leading to the deformation of the CNTs and cessation of emission[17]. The emission is then substituted by other CNTs, which leads the emitting current to fluctuate and the system to fail prematurely once the majority of CNTs are deformed[19,20]. For another type of cathode, the gate-type cathode is considered as the suitable neutralizer for EP systems such as the one on the LISA-PATHFINDER

[1]School of Optics and Electronic Information, Huazhong University of Science and Technology, Wuhan 430074 Hubei, China. [2]MOE Key Laboratory of Fundamental Physical Quantities Measurement & Hubei Key Laboratory of Gravitation and Quantum Physics, PGMF and School of Physics, Huazhong University of Science and Technology, Wuhan 430074 Hubei, China. [3]Hubei Key Laboratory of Plasma Chemistry and Advanced Materials, School of Materials Science and Engineering, Wuhan Institute of Technology, Wuhan 430205 Hubei, China. [4]MOE Key Laboratory of TianQin Mission, TianQin Research Center for Gravitational Physics & School of Physics and Astronomy, Frontiers Science Center for TianQin, Gravitational Wave Research Center of CNSA, Sun Yat-sen University (Zhuhai Campus), Zhuhai 519082, China. [5]Lab of Plasma Propulsion, Harbin Institute of Technology (HIT), Harbin 150001, China. ✉e-mail: sunleimeng@hust.edu.cn; SONGPEIYI@hust.edu.cn

satellite, since the overall uniform gate structure guarantees the uniform electric field strength on the CNTs[21]. But the E-beam is inevitably intercepted by the gate, which reduces the transmittance (less than 70%)[22]. The E-beam will collide with the gate to cause structural deformation and short-circuits, ultimately leading to electric field non-uniformity and CNTs damages[23,24].

Various methods have been investigated for increasing the transmittance of gate-type cathode[21,25]. The most straightforward approach is to reduce the gate aperture ratio, whereas the thickness and width of the gate cannot be too small in order to meet the structural robustness and material processing requirements[26]. Alternative approaches, by placing external forces such as electrostatic and magnetic forces on the E-beam, are considered to achieve the E-beam focusing for higher transmittance[27,28]. Han et al. developed a curve-shaped elliptical focusing lens above the gate, which achieved a high E-beam transmittance of 95.2%, showing a path for the high-performance X-ray tube[29]. Zhang et al. demonstrated the effectiveness of introducing an external magnetic field to enhance the performances of the gate-type cathodes. By implementing a periodic permanent magnet (PPM) focusing field, a maximum collector current of 7.52 mA with the corresponding transmittance of 93.07% was achieved[30]. Although these methods demonstrate the proof-of-concept, the size, complexity, and power consumption of the system are increased, and the effectiveness of these methods is highly dependent on the precise vertically alignment between the set-up and the beam line, which is difficult to achieve and maintain[31].

In this work, we present an approach that the E-beam focusing in gate-type cathodes can be achieved simply using a self-charging gate. Specifically, an electret material-based gate with $SiN_x$/Au/Si structure is introduced. Notably, this configuration enables ultrahigh transmittance (~96%) without any external instrument. Upon the cathode turns on, the electrons emitted from the CNTs are effectively trapped by the Si hanging bonds in the $SiN_x$, resulting in the negative charging on the gate surface[32,33]. The surface induces a local electric field that generates electrostatic forces on subsequently emitted electrons. This force causes the deflection of flying electrons to let them pass through the gate with minimal interceptions. Owing to the material structure of the plasma-enhanced chemical vapor deposition (PECVD) grown $SiN_x$,

the gate exhibits a high surface charge density after charging. The surface potential ($V_s$) of the $SiN_x$/Au/Si gate is measured to be approximately −60 V after charging the cathode continuously for 12 h, verifying the exceptional charge storage capacity of $SiN_x$. The mechanism is verified by our numerical simulations of the electron trajectories, and the simulation also confirms that the introduction of the $SiN_x$ electret gate does not compromise the electric field uniformity of the emitters. Our cathode prototype based on the design is monitored continuously for 550 h, with a mean current of 26.51 μA and a mean fluctuation of 8.2%. The mean E-beam transmittance is 96.17% with a fluctuation of less than 1.0% throughout the entire testing, indicating the high transmittance and long-term stability of the cathode.

## Results
### Device design and operating principle
To reduce the gate current loss and improve the E-beam transmittance, we developed the $SiN_x$/Au/Si gate to be incorporated with the CNTs-based cathode. The operating principle of the proposed cathode, which defined by the $SiN_x$ electret-based gate, is illustrated in Fig. 1a. In general, the cathode operates through three main steps: electrons emission, electrons injection, and electrons focusing. It is widely acknowledged that appropriate surface field strengths on CNTs emitter can significantly modify the potential barrier and induce field emission (FE) of electrons[34]. The emitted electrons fly towards the gate under the electric field force. Unlike the commonly used metal-based gate, which immediately releases the electrons to ground upon incidence, the proposed $SiN_x$/Au/Si gate can trap and store the colliding electrons due to the interface traps of $SiN_x$ electret[32]. Next, the electrons injection of the $SiN_x$ electret will introduce the negative $V_s$ of the gate, which creates a local electric field between gate units. Before the electrons injection reaches saturation, only a portion of the electrons can pass through the gate, while the others are intercepted, resulting in the E-beam loss. Once after enough electrons are injected onto the $SiN_x$ surface, the trajectories of subsequent electrons will be altered to prevent their interaction with the gate. Finally, the majority of E-beam are focused to pass through the gate, leading to a high transmittance of the cathode[35].

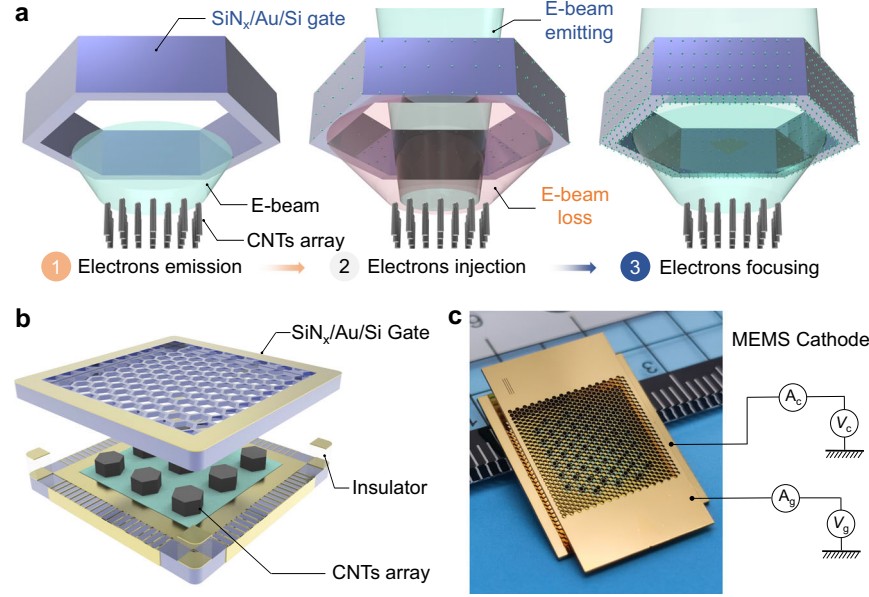

**Fig. 1 | Structure design overview of the $SiN_x$-based cathode. a** Operating principle of the proposed cathode based on $SiN_x$/Au/Si gate, the electron beam (E-beam) emitted from carbon nanotubes (CNTs) array can be focused after the charging of the $SiN_x$/Au/Si gate. **b** Schematic of the cathode composed of the gate, insulator, and CNTs emitter. **c** The optical image of the micro-electromechanical systems (MEMS) cathode with dimensions of 15 × 25 × 1.5 mm (length × width × height). The currents can be measured using the electrometers ($A_c$ and $A_g$) under applied emitter voltage ($V_c$) and gate voltage ($V_g$).

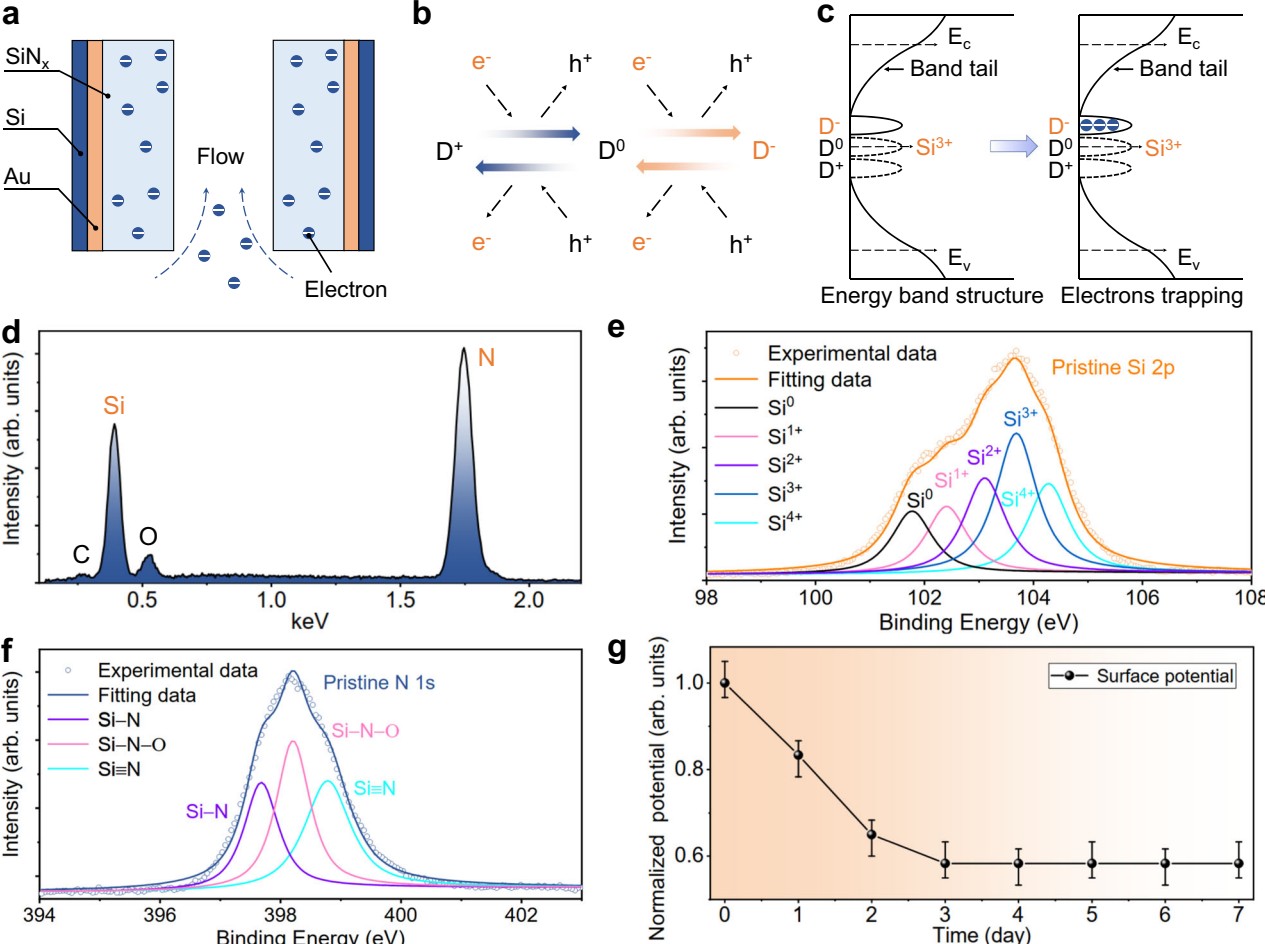

**Fig. 2 | Electron trapping model and material characterizations of SiNx electret.**
**a** Self-charging of the SiNx/Au/Si gate using electron beam (E-beam) irradiation.
**b** Conversion model for the three charges states due to the interaction with electrons (e⁻) and holes (h⁺). When a single electron is captured by a dangling bond, the charge state is neutral ($D^0$). The negative ($D^-$) state occurs when two electrons are attached to the dangling bond. Conversely, in the absence of electron bonding, a hole attaches to the dangling bond, resulting in a positive ($D^+$) state. **c** The band structures of SiNx before and after electrons trapping, showing defect states ($Si^{3+}$), band-tails, conduction band ($E_c$) and valence band ($E_v$). **d** The energy dispersive X-ray (EDX) spectrum of the SiNx film. **e** Si 2p X-ray photoelectron spectroscopy

(XPS) spectra for the SiNx film. The yellow circle and curve represent the original Si 2p peak and fitted Si 2p peak, respectively, while the black, pink, purple, dark blue and navy curves correspond to the peaks of $Si^0$, $Si^{1+}$, $Si^{2+}$, $Si^{3+}$, and $Si^{4+}$ after fitting respectively. **f** N 1s XPS spectra for the SiNx film. The dark blue circle and curve represent the original N 1s peak and fitted N 1s peak, respectively, while the purple, pink and navy curves correspond to the peaks of Si–N, Si–N–O and Si≡N after fitting respectively. **g** Normalized surface potential decay versus time. The error bars correspond to the largest relative deviation of the peak-to-peak potential from the mean measurements. Source data are provided as a Source Data file.

Figure 1b shows the schematic of the proposed cathode, which consists of the SiNx/Au/Si gate, insulator, and CNTs emitter. As mentioned previously, the electric field uniformity of cathode emitters can be improved by decreasing the mesh size of the gate. To achieve this, we utilized the CST simulation software to optimize the gate structure while taking into considerations micro-fabrication conditions. Ultimately, a 300 μm thick gate with regular hexagon shapes measuring 228 μm in side length and 20 μm in separation is designed for electrons extraction. As outlined in Supplementary Fig. 1 and Note 1, the Si-based gate was fabricated using a deep reactive ion etching (DRIE) process, followed by deposition of a 400 nm thick Au electrode using the E-beam evaporation (EBE) technique. To achieve the electron focusing, an electret layer (SiNx) with a thickness of 1 μm was deposited onto the Au/Si gate using the PECVD technique. The investigation of the stoichiometry of the SiNx electret can be found in Supplementary Fig. 2 and Note 2[36,37]. To enhance the emission performances of the cathode, the patterned CNTs were used as emitters and transferred onto the metal substrate using thermocompression processes technology. This patterning transfer method improves the adhesion between CNTs and the target substrate while simultaneously

weakening the shielding effects[38,39]. Further details regarding the transfer process and shielding effects can be found in Supplementary Fig. 3 and Note 3. After the fabrication of the gate and CNTs emitter, the cathode device was assembled using metal-metal bonding as illustrated in Supplementary Fig. 1. Finally, the cathode device, based upon SiNx/Au/Si gate and patterned CNTs emitter, was successfully obtained as demonstrated in Fig. 1c. Further morphology characterizations, including optical and scanning electron microscope (SEM) images, can be found in Supplementary Fig. 4 and Note 4.

## Characterizations of the SiNx electret

An improved understanding of the electron trapping mechanism is crucial for developing a comprehensive model of the SiNx/Au/Si gate. The foundational model of electret for charge storage is based on electret-metal structure, in which the incident charged particles (ions or electrons) can be stored in the electret material to form the positive or negative $V_s$, leading to the charge compensation of opposite polarity in the metal electrode[40]. The detailed charge storage model of the SiNx/Au/Si gate can be found in Supplementary Figs. 5, 6 and Supplementary Note 5. In the SiNx/Au/Si gate structure depicted in Fig. 2a, electrons are

stored in the SiN$_x$ electret through the E-beam irradiation, which is a commonly used method for charge injection.

To be more precise, the amphoteric traps in SiN$_x$ electret, which are caused by the dangling bonds, are responsible for the surface charging[41]. In this model, both electron and hole charge states are charged attributed to the trivalent Si (Si$^{3+}$) center. This center consists of a Si atom with a dangling bond and three other bonds satisfied by three nitrogen atoms[33,42]. As illustrated in Fig. 2b, when a single electron is trapped by the dangling bond, the charge state is neutral (D$^0$), and the negative (D$^-$) state is formed when two electrons are attached to the dangling bond. Conversely, in the absence of electron bonding, a hole is attached to the dangling bond, resulting in a positive (D$^+$) state. The detailed states conversion between these three charges can be described through four capture processes and four emission processes of electrons and holes. The D$^-$ state, corresponding to the negative $V_s$, can be employed to explain the electrons trapping in the SiN$_x$/Au/Si gate effectively. Figure 2c illustrates the band structures of SiN$_x$ before and after electron capturing, showing both defect states (Si$^{3+}$ centers) and band-tails (details can be found in Supplementary Note 6). The defect states can be represented as D$^+$, D$^-$, and D$^0$ states depending on the number of electrons captured by the dangling bonds. When two electrons are captured by the dangling bonds, the band structure of SiN$_x$, as shown in Fig. 2c, indicates that the trap levels generated by Si$^{3+}$ centers are occupied by electrons, resulting in a negative surface potential of the SiN$_x$. More details for the trap distributions under different deposition conditions of SiN$_x$ can be found in Supplementary Fig. 7 and Note 7.

The material characteristics of SiN$_x$ electret are also essential for understanding the electron trapping mechanism. To characterize the properties of SiN$_x$ electret film in detail, elemental analysis was conducted using an energy dispersive X-ray (EDX) spectra system to determine the elemental compositions of SiN$_x$ before the FE process. As shown in Fig. 2d, the main components are Si and N, along with trace amounts of C and O elements. This phenomenon can be attributed to the formation of silicon oxide due to the oxidation of Si atoms in the atmosphere during the preparation of SiN$_x$ film or the presence of C and O elements in the air. Furthermore, research on the impact of hydrogen outgassing on the SiN$_x$ electret can be found in Supplementary Note 8.

To determine the chemical structure of the SiN$_x$ film deposited by PECVD, X-ray photoelectron spectroscopy (XPS) analysis was performed using a large 500 mm Rowland circle monochromated Al Kα X-ray source. Results show that the chemical structure of SiN$_x$ film is a mixture of all the silicon nitride states, which depends on the arrival types of the N and Si atoms and their combination on the gate surface during the deposition. The mixture of silicon nitride states results in a broad XPS peak around 103 eV, and the Si 2p and N 1s binding energies revealed from XPS analysis of SiN$_x$ film are indicated in Fig. 2e and Fig. 2f, respectively. Concerning the chemical structure of amorphous SiN$_x$ film, the Si 2p spectra exhibit the multi-peak behavior that can be de-convoluted into five distinct peaks, which can be explained using the random bonding model proposed in previous work[43]. According to the random binding mode, the Si 2p line is the result of the superposition of the five [Si−(Si$_n$N$_{4-n}$), $(0 \le n \le 4)$] components which follow a statistical distribution[44]. As the value of n increases, the peak energy of Si−(Si$_n$N$_{4-n}$) shifts from around 101.8 eV to 104.3 eV. Figure 2e shows the five fitted peaks of Si$^0$ (101.8 eV), Si$^{1+}$ (102.4 eV), Si$^{2+}$ (103.1 eV), Si$^{3+}$ (103.7 eV), and Si$^{4+}$ (104.3 eV) obtained through the de-convolution of the Si 2p spectra, corresponding to zero, one, two, three or four Si atoms of the tetrahedral Si−Si bonds are replaced by the Si−N bonds. The appearance of binding energies at 101.8 eV and 102.4 eV in the fabricated sample suggests the formation of SiN$_x$:H in the SiN$_x$ film, along with other binding energies corresponding to the generation of SiO$_x$ and SiO$_x$N$_y$, as reported in earlier studies[45–47]. Additionally, the significant amounts of Si$^{3+}$ binding energy further explain the

performance for electrons trapping of SiN$_x$ electret discussed earlier. Moreover, the N 1s phase is mainly composed of Si−N (397.7 eV), Si−N−O (398.2 eV) and Si≡N (398.8 eV) from the XPS analysis seen in Fig. 2f[48]. Most importantly, the portion chemical bonds of silicon nitride mixture such as Si−N, Si−H and N−H bonds might be broken and transform into the Si and N dangling bonds under ion bombardment during the PECVD process[49,50], and thus the electrons will be trapped by Si dangling bonds in the SiN$_x$ film after the E-beam irradiation.

To confirm the electron trapping capability of the SiN$_x$ film, we investigated the $V_s$ of the proposed gate using a capacitance electrometer (Model EST102). As indicated in Fig. 2g, the $V_s$ of the gate based on SiN$_x$ electret is approximately -60 V after E-beam irradiation for 12 h, which corroborates the theoretical analysis. Whereas the $V_s$ shown in Fig. 2g decreases by around 40% in a week in the atmosphere, attributed to the formation of the water layer on the gate surface. The detailed descriptions for surface charge decay of the SiN$_x$ electret can be found in Supplementary Fig. 8 and Note 9[51].

## Modeling and simulation of cathodes with different gates

To elucidate the reason underlying the high E-beam transmittance, we performed the electric field and E-beam trajectory simulations using model geometries of the cathodes with different gates, mimicking those used in the experiments. Specifically, in the case of the cathode with an Au/Si gate, the portion emitted electrons from the CNTs are immediately released due to the conductivity of metal, which causes E-beam divergence and results in the low E-beam transmittance observed in Fig. 3a. In contrast to the Au/Si gate, the SiN$_x$/Au/Si gate (Fig. 3b) captures the incident electrons rapidly, creating a negative $V_s$ that significantly alters the trajectories of subsequent electrons, resulting in E-beam focusing and a higher E-beam transmittance. To further illustrate this effect, another simulation is used to investigate the electric field distributions of different cathodes. The schematic of the FE triodes setup is indicated in Fig. 3c, with a distance of 0.5 mm between the CNTs emitter and the gate. The potentials of the anode ($V_a$), the Au-gate ($V_g$) and the CNTs emitter ($V_c$) are set to 0 V, 0 V and −500 V, respectively. The simulation results (Fig. 3c) show that the electric field is highly concentrated between the Au-gate and the CNTs emitter. The enlarged view of the potential distribution around the Au-gate suggests that the emitted electrons move along the tangent direction of the potential line and eventually be released, resulting in a current loss.

On the other hand, the electric field distribution of the cathode with the SiN$_x$-gate is also investigated, as shown in Fig. 3d, with the electric potential of the cathode set to the same value as that in Fig. 3c, and the negative potential (−60 V) of SiN$_x$ material is incorporated in the simulation. The simulation results reveal that the electric field is concentrated on the overall cathode and is significantly enhanced between the SiN$_x$-gate and anode, as a result of the introduced negative potential. As shown in the enlarged drawing in Fig. 3d, the emitted electrons can be focused instead of being released under the convergent electric field force and will quickly arrive at the anode due to the enhanced electric field, leading to a high E-beam transmittance. As previously stated, the electric field uniformity of the CNTs emitter is crucial for the long-term stability of the cathode, and the results presented in Supplementary Figs. 9, 10 and Supplementary Note 10 indicate that the electric field distributions within the CNTs emitter are uniform, with minimal variations in electric field strength observed both before and after the introduction of the SiN$_x$ electret.

In addition, the geometry and dimension of the cathode components, including the CNTs emitter and the gate electrode, were established and simulated using the CST STUDIO SUITE. The emission currents and the electron trajectories are calculated. Figure 3e, f show the cross-sections and electron trajectories corresponding to the cathode with the Au-gate and SiN$_x$-gate, respectively. The E-beam

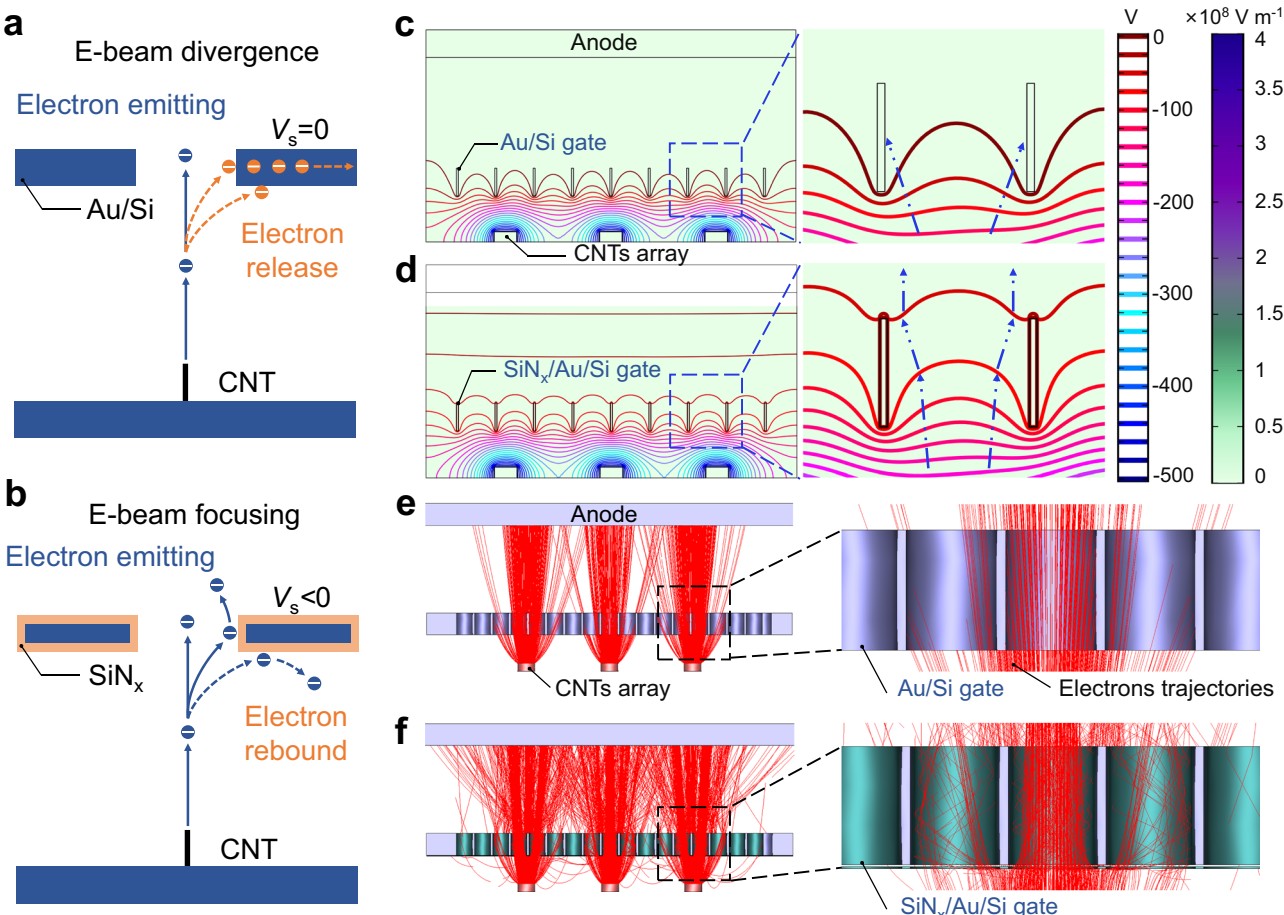

**Fig. 3 | Simulation results of potential distributions and electrons trajectories of the cathodes with different gates. a** Schematic of electron beam (E-beam) divergence based on a single carbon nanotube (CNT) and the Au-gate, the surface potential ($V_s$) of the Au-gate is equal to 0. **b** Schematic of E-beam focusing based on a single CNT and the SiN$_x$-gate, the $V_s$ of the SiN$_x$-gate is less than 0. The potential distributions between the carbon nanotubes (CNTs) emitter and the anode in the cathode equipped with (**c**) Au-gate and (**d**) SiN$_x$-gate, respectively. A side-view cross-section of E-beam trajectories from the CNTs emitter to the anode in the cathode equipped with (**e**) Au-gate and (**f**) SiN$_x$-gate, respectively.

originates from the CNTs surface, passes through the gate, and finally reaches the anode, with the current values calculated accordingly. The cathode current density of the FE simulation model is given by the Fowler-Nordheim (F-N) expression[52]:

$$J = aE^2 e^{-\frac{b}{E}} \quad (1)$$

where $J$ is the current density of the cathode, $E$ represents the electric field intensity. $a$ and $b$ are the FE linear factor and index factor, respectively, which are related to the surface shape and work function of the emitter materials. The values of $a = 4.75 \times 10^{-10}$ A V$^{-2}$, $b = 2.54 \times 10^6$ V m$^{-1}$ can be obtained by fitting the experimental results using data processing software. In the simulation model, the E-beam transmittance of the cathode with different gate structures can be obtained by calculating the gate current ($I_g$) and anode current ($I_a$). The cathode with SiN$_x$-gate demonstrates >97% transmittance (equal to $I_a$/($I_g + I_a$)) under applied $V_c$, which is crucial for enhancing the operating stability and lifespan of the FEC. We further investigated the E-beam trajectories of the different cathodes and found that due to the E-beam divergence, a significant number of electrons are released instead of passing through the Au-gate, resulting low transmittance, as depicted in Fig. 3e. However, the simulation results show that when the SiN$_x$-gate is used, the E-beam trajectories are concentrated, as shown in Fig. 3f, resulting in a substantially lower interception rate and higher transmittance.

## The operation performances of the cathode with SiN$_x$-gate

To validate the simulation results, we quantitatively characterized the electron emission properties of the proposed cathode based on the SiN$_x$ electret by measuring the emission current in a triode configuration, as shown in the insert of Fig. 4a. The experimental sample for the FE measurement was placed in a vacuum chamber at $1 \times 10^{-6}$ Pa. The FE process was carried out using the Wisman DL5N300 high voltage source, and the Keithley 6514 electrometer, with high accuracy and sensitivity, was used for current acquisition, including $I_a$ and $I_g$, and cathode current $I_c = I_a + I_g$. The detailed current measurement system can be found in Supplementary Fig. 11 and Note 11. According to the F-N theory, the FE current $I$ can be described as follows[53]:

$$I = A\alpha \left(\frac{\beta^2 E^2}{\varphi}\right) \exp\left(-\frac{B\varphi^{3/2}}{\beta E}\right) \quad (2)$$

where $\alpha$, $\beta$ and $\varphi$ are the effective emission area, field enhancement factor and work function (5.0 eV) of CNTs emitter, respectively. A and B are constant values of $1.54 \times 10^{-6}$ (A V$^{-2}$ eV) and $6.83 \times 10^9$ (eV$^{-3/2}$ V m$^{-1}$), respectively. As demonstrated in Fig. 4a, under an appropriate $V_c$, the currents can be simultaneously collected using a high-sampling-rate (2000 Hz) acquisition system immediately after the cathode's initiation, and the E-beam can be calculated as well. The test results indicate that at the moment of cathode startup,

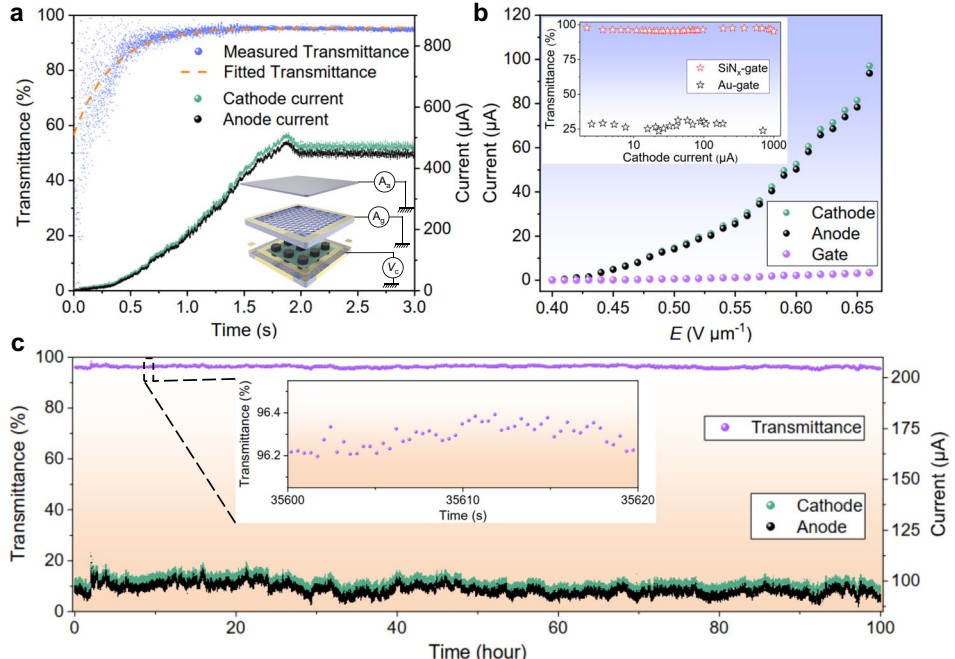

**Fig. 4 | Characterizations of the operation performances of the proposed cathode. a** The emission currents and transmittance as functions of the turn-on time of the cathode. The insert is the measurement schematic of the triode structure, the currents of anode and gate can be measured using the electrometer ($A_a$ and $A_g$) under applied emitter voltage ($V_c$). The blue, green, and black dots correspond to the measured electron beam (E-beam) transmittance, cathode current, and anode current data, respectively. Meanwhile, the yellow dashed line represents the fitted E-beam transmittance. **b** Current-electric field (*I-E*) curves of the cathode with SiN$_x$-gate, the inset is a comparison of E-beam transmittances between cathodes based on SiN$_x$-gate and Au-gate. The green, black and purple dots correspond to the measured cathode current, anode current and gate current, respectively. Meanwhile, the red and black five-pointed stars in the inset represent the measured E-beam transmittances based on SiN$_x$-gate and Au-gate, respectively. **c** The 100 h current stability of the cathode upon -100 µA, the inset is an enlargement of the E-beam transmittance over a 20 s interval. The emitter area is 0.057 cm². The purple, green and black dots correspond to the measured E-beam transmittance, cathode current and anode current, respectively. Source data are provided as a Source Data file.

the emission current is minimal, resulting in a low and unstable E-beam transmittance through the corresponding SiN$_x$-gate. As the cathode emission current gradually increases, the charging of the SiN$_x$-gate saturates over time, leading to a gradual increase in the E-beam transmittance through the gate, eventually reaching a stable state. The measured and fitted curves of the E-beam transmittance in Fig. 4a demonstrate that the time required for the gate E-beam transmittance to exceed 90% is less than 1 s. Furthermore, it takes less than 1.5 s for the E-beam transmittance through the gate to reach its maximum value of 95.50%. The results in Supplementary Fig. 12 and Note 11 also show that the cathode can realize high E-beam transmittances in a brief period under different current levels.

Upon achieving the charging saturation of the SiN$_x$-gate, increasing the voltage on the cathode to improve the cathode emission current/current density should, theoretically, not alter the surface potential of the SiN$_x$-gate. This phenomenon occurs due to the electron trapped within the SiN$_x$ have reached saturation at this stage, resulting in a stable surface potential[54]. Details can be found in Supplementary Figs. 13, 14 and Note 12. Based on this premise, the cathode's emission performance and transmittance over a wider current range were investigated. We measured the current-electric field (*I-E*) curve, as shown in Fig. 4b. The turn-on electric field $E_{to}$ (corresponding to cathode current density of 10 µA cm$^{-2}$) and threshold electric field $E_{th}$ (corresponding to cathode current density of 1 mA cm$^{-2}$) can be obtained from the *I-E* curve and are found to be about 0.41 V µm$^{-1}$ and 0.61 V µm$^{-1}$, respectively[55,56]. Additionally, F-N plots can be calculated based on the measured emission currents (Supplementary Fig. 15 and Note 13), revealing that the electrons emission from the CNTs emitter follows the features of quantum tunneling behavior. From the F-N

plots presented in Supplementary Fig. 15, the $\beta$ of CNTs emitter is calculated to be 14,850 using the following equation[2]:

$$\beta = -B\varphi^{3/2}d/S \tag{3}$$

in which B is a constant value from Eq. (2), d indicates the distance between the gate and CNTs emitter in the triode structure, and $S$ represents the slope of the F-N plot. The high $\beta$ demonstrates the good electric field uniformity on CNTs emitter surface as well.

As shown in Fig. 4b, the average E-beam transmittance remains high at 96.05% within about 0-100 µA range (corresponding to a current density of 0–1.75 mA cm$^{-2}$) according to the FE curves, indicating that the SiN$_x$-based gate transmittance is robust under varying emission currents. To validate the E-beam transmittance of the cathode at higher current levels, following the saturation of charging in SiN$_x$-gate, we conducted measurements of E-beam transmittances through the SiN$_x$-gate for cathode emission current ranging from 2.14 µA to 1 mA (corresponding to the emission current density of 0.037 mA cm$^{-2}$ to 17.54 mA cm$^{-2}$). As depicted in the inset of Fig. 4b, the test results indicate that during the process of increasing the emission current/current density by enhancing the cathode voltage, the E-beam transmittance through the SiN$_x$-gate remains insignificantly altered. The average E-beam transmittance across the gate from 2.14 µA to 1 mA is measured at 96.33%. In contrast, when the FE performances of the Au-gate were evaluated at various emission currents using the same experimental setup, the average transmittance drops to 27.21% within the 2.50 µA$^{-1}$ mA range due to the Au-gate interception and space-charge effect during the FE process depicted in the inset of Fig. 4b[57]. The average E-beam transmittance of the SiN$_x$-based cathode is increased by 254% to about 96.33% at

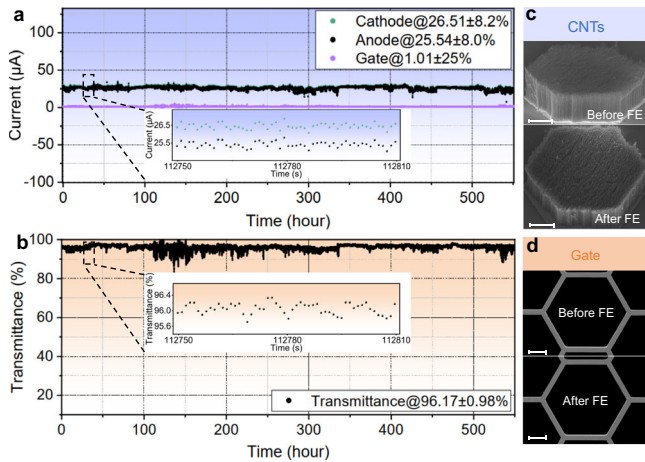

**Fig. 5 | Long-term current measurements of the cathode with SiN$_x$-gate.**
**a** Emission current stability of the cathode, the inset illustrates the current fluctuations of cathode current and anode current within 1 min. The green, black and purple dots correspond to the measured cathode current, anode current and gate current, respectively. **b** Transmittance of the cathode, the inset is the transmittance fluctuation within 1 min. The black dot represents the measured E-beam transmittance based on SiN$_x$-gate. **c** Surface morphologies of the carbon nanotubes (CNTs) emitter before and after long-term field emission (FE). Scale bars, 100 μm. **d** Surface morphologies of the SiN$_x$-gate before and after long-term FE. The emitter area is 0.08 cm². Scale bars, 100 μm. Source data are provided as a Source Data file.

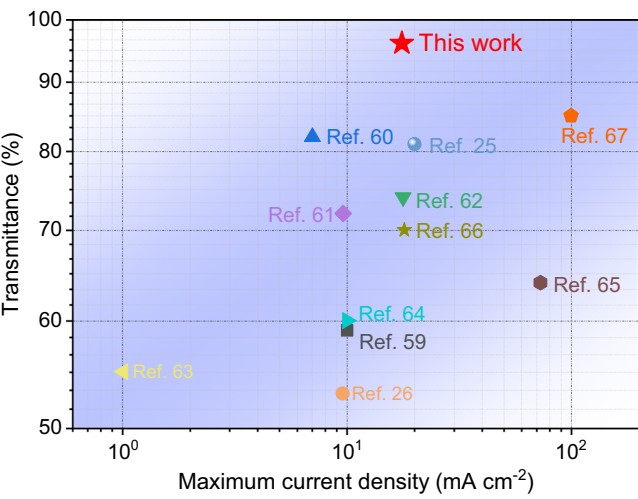

**Fig. 6 | The performances comparison of the gate-type cathodes.** Comparison of performances of the proposed cathode among various gate-type cathodes based on carbon nanotubes (CNTs) emitter. Source data are provided as a Source Data file.

about 0-1 mA current range owing to the electret material's electrostatic focusing effect.

Simultaneously, we investigated the thickness of the SiN$_x$ electret effect on the E-beam transmittance of the cathode. Based on the research findings, when the thickness of SiN$_x$ is less than 1 μm, the surface potential of the SiN$_x$-gate after charging is relatively low, leading to poor electrons focusing and lower transmittance. When the thickness is equal to or greater than 1 μm, the surface potential saturates, enabling the SiN$_x$-gate to achieve ultrahigh E-beam transmittance over a wide range of currents. Consequently, we have chosen a SiN$_x$ thickness of 1 μm for our final selection. For more detailed information, please refer to Supplementary Fig. 16 and Note 14. Moreover, to verify the long-term stability of the cathode based on a SiN$_x$-gate under high current levels (~100 μA), we conducted a prolonged stability test for

cathode emission. As illustrated in Fig. 4c, during a continuous 100 h FE test, the cathode consistently maintains an average $I_c$ of 99.40 μA, with a minimal current fluctuation of 2.5%. The mean $I_a$ is 95.69 μA, with a minimal current fluctuation of 2.8%. Furthermore, the average E-beam transmittance through the SiN$_x$-gate is measured at 96.25%, exhibiting a minor transmittance fluctuation of 0.42%. Throughout the entire testing process, there is no noticeable degradation or fluctuation in the E-beam transmittance of the SiN$_x$-gate, indicating that the SiN$_x$-gate's electron-focusing effect remains constant.

For missions like deep space exploration, where long-time operation up to a few years is required, the cathode's long-term stability and high transmittance are crucial. To evaluate the long-term stability of our cathode, a vacuum chamber was used for conducting the aging process without interrupting the measurement setup. The cathode prototype with SiN$_x$-gate is operated at a fixed emission current of approximately 27 μA with the closed-loop control. Figure 5a illustrates the emission current over 550 h, with an average $I_c$ of 26.51 μA (corresponding to the current density of 0.33 mA cm⁻²) and current fluctuation of 8.2%, and the insert in Fig. 5a shows an enlarged view of $I_c$ and $I_a$ within 1 min.

From the measured current data, the transmittance of the cathode is obtained shown in Fig. 5b. The results demonstrate a mean transmittance of 96.17% with a fluctuation of 0.98% during the 550 h test, confirming the electret's stability for electron capture and maintaining. The insert in Fig. 5b provides an enlarged view of E-beam transmittance within 1 minute. After 550 h of operation, SEM images of the CNTs emitter and gate (Fig. 5c, d) show that the surface morphologies of the cathode remain intact, indicating the uniform electric field on the CNTs and low gate current loss, which facilitates the stable emission of the cathode. Additionally, it should be noted that the initial $V_c$ value at the onset of the experiment is 517.2 V. Subsequently, during the 550 h test, the voltage exhibits a gradual decrease, which is attributed to the FE process stabilizing during the extended duration of the measurement[58]. The average rate of voltage decline during the testing period is 56.4 mV hr⁻¹. For additional information, please refer to Supplementary Fig. 17 and Note 15. To verify the long-term stability of CNTs FEC based on a SiN$_x$-gate under high current (~mA level), we carried out a long-term test (60 hours) for cathode emission at around 3.25 mA (corresponding to a current density of 9.50 mA cm⁻²). The average E-beam transmittance through the SiN$_x$-gate is measured to be 95.30%, with a transmittance fluctuation of 0.30%. For specific details, please consult the data provided in Supplementary Fig. 18 and Note 16. Moreover, the E-beam transmittance and long-term stability tests (310 h) of the cathode engineering prototype upon the external stimuli were also conducted to verify the stability and reliability of the charging of SiN$_x$-gate, and test results indicate that the cathode can maintain current stability and high E-beam transmittance even under/after external stimulation (Supplementary Figs. 19–31 and Supplementary Notes 17, 18). Therefore, the as-prepared cathode with long-term stability and high transmittance under wide-range current and external stimuli is well-suited for the in-space applications mentioned above.

The performances of the SiN$_x$-gate cathode are evaluated by comparing the maximum current density and the corresponding transmittance to other gate-type cathodes based on CNTs emitter (Fig. 6)[25,26,59–67]. The comparison results demonstrate that the SiN$_x$-gate based cathode exhibits a significantly higher transmittance. Most gate-type cathodes based on CNTs exhibit E-beam transmittance levels ranging from 50% to 80%. In contrast, our designed FEC achieves an average E-beam transmittance of 96.33% at the current density level ranging from 0.037 mA cm⁻² to 17.54 mA cm⁻², significantly reducing cathode power consumption and enhancing the stability of the FEC during long-term operation. By the way, it is noteworthy that certain cathodes exhibit a greater current density magnitude compared to the cathode we have designed. This phenomenon can be attributed to the smaller emission areas of CNTs and higher applied voltages in some cathodes. It is possible to enhance the current of our cathode by

simply increasing its effective CNTs area without reducing the transmittance. Moreover, the ultrahigh transmittance of the gate not only reduces the power consumption but also contributes to the long-term stable operation of the cathode system, which is a key factor for the cathode to be successfully used in space missions.

## Discussion

We present a method to achieve the high E-beam transmittance of gate-type FEC by using a SiN$_x$ electret material-based gate. The Si hanging bonds in the SiN$_x$ provide positions for trapping the electrons, making the gate surface negatively charged. The surface potential of SiN$_x$ is measured to be about −60 V after the cathode turning on, which generates a strong electrostatic force to focus the E-beam, the transmittance reaches its maximum in a few seconds and maintains stable in continuous emission. The phenomenon is verified by both numerical simulation and device characterizations. The prototype developed based on this idea demonstrates that the mean transmittance is 96.17% with a fluctuation of less than 1.0% throughout the 550 h of testing. The transmittance above 95% is recorded for the cathode current from 2.14 μA to 3.25 mA, producing the current density up to 17.54 mA cm$^{-2}$. The cathode has a simple structure and can be fabricated using the standard micromachining processes, and the test prototypes exhibit reliable performances upon external stimuli such as impact, vibration, thermal, light and electrical stimulation. To our best knowledge, this is the highest transmittance achieved on the gate-type cathodes reported to date without sacrificing the FE efficiency of CNTs. We believe that this high-efficiency, long-lifetime cathode will be an ideal electron source for electric thrusters, solar sails and tethered satellites in long-term space missions.

## Methods

### Fabrication of the SiN$_x$/Au/Si gate

The SiN$_x$/Au/Si gate was fabricated using a lithographic patterning process and film deposition techniques. Initially, a direct-write lithographic process was employed to pattern a 13 μm AZ9260 photoresist layer as an etch-mask on a 300 μm Si wafer, resulting in etching grooves after development. The wafer was then subjected to a DRIE process to create the final hexagonal gate structure with a backside aluminum stop layer. Subsequently, the residual photoresist on the gate structure was removed by organic solvent and oxygen plasma in sequence. The backside aluminum stop layer was etched using an alkali solution, and the sample was thoroughly rinsed with deionized water and dried with nitrogen flow to obtain the gate framework. Finally, a 400/50 nm thick Au/Ti electrode was deposited by the EBE process, and a SiN$_x$ film with 1 μm thick was deposited on the Au/Si gate using the PECVD process. During the deposition of the SiN$_x$ film, the flow rates of SiN$_4$, NH$_3$, and N$_2$ were set at 15 sccm, 11 sccm, and 785 sccm, respectively. The high frequency (HF) and low frequency (LF) were set to 13.5 MHz and 697 kHz, with corresponding deposition times of 13 s and 7 s per cycle, respectively. The RF power was set to 50 W, and the substrate temperature was maintained at 100 °C.

### Fabrication of the carbon nanotubes emitter

The CNTs emitter on the Si wafer was synthesized via a thermal chemical vapor deposition (TCVD) system. Firstly, a 15 nm Al$_2$O$_3$ layer and a 1 nm Fe catalyst layer were sequentially deposited on a 4-inch Si wafer using an atomic layer deposition (ALD) system. The Al$_2$O$_3$ layer served as the buffer layer while the Fe layer acted as the catalyst. Subsequently, under inert atmospheres (Ar flow with 1300 sccm H$_2$), the CNTs array was grown at 750 °C, with a carbon source supplied by C$_2$H$_4$ at a flow rate of 1000 sccm. Finally, the chamber was cooled down to room temperature while maintaining an Ar atmosphere. Before the transfer process, the CNTs were heated in the atmosphere at 500 °C for 20 min and deposited with Au/Ni/Cr (500/20/50 nm) films on the top side. During the transfer process, the CNTs on the Si wafer and another target substrate with patterned Au/Ni/Cr (500/20/50 nm) on the surface were aligned and placed on the mechanical platform. After applying pressure of 30 kPa and temperature (300 °C) on both platforms simultaneously for 5 min, the CNTs and target substrate were bonded together by the alloy. Subsequently, the patterned CNTs emitter was obtained after removing the original substrate. Finally, the CNTs array with a thickness of approximately 110 μm, regular hexagon shapes of 240 μm side, and 900 μm separation on the target substrate was obtained as the cathode emitters.

### Preparation of the cathode

The CNTs emitter was carefully aligned and assembled with the SiN$_x$/Au/Si gate through a separator layer. Firstly, a 400/50 nm thick Au/Ti electrode was deposited on both the top and bottom of the glass substrate using the EBE technique. Subsequently, the CNTs emitter and the SiN$_x$/Au/Si gate were aligned in sequence and bonded using thermocompression technology through the separator layer. During the bonding procedure of the CNTs emitter and the gate, we utilized the flip-chip bonder (FINEPLACER® sigma), which offers sub-micron placement accuracy, ensuring precise alignment between the gate hole and CNTs emitter. Finally, the cathode was successfully assembled with high precision and reliability.

### Characterizations of the cathode

The cathode morphologies including the CNTs emitter and gate were observed using an optical microscope (OLYMPUS BX51M) and a SEM (ZEISS Sigma VP). The EDX microanalysis as a non-destructive analytical technique was used to identify and characterize the elemental composition of the SiN$_x$ film. The XPS (AXIS Supra$^+$) with a large 500 mm Rowland circle monochromated Al Kα X-ray source and optimized electron optics ensuring chemical resolution was used to determine the chemical structure of the SiN$_x$ film. Additionally, the $V_s$ of charged SiN$_x$ film was measured using the EST102 capacitance electrometer with a maximum measurement range of 20 kV.

### The field emission properties measurements of the cathode

The FE characteristics of the cathode were tested in a vacuum chamber with a pressure level of $1 \times 10^{-6}$ Pa. The FE currents were measured using a Keithley 6514 electrometer under a high negative voltage applied by a Wisman DL5N300 power supply, in a triode configuration. The gate-to-emitters distance is 500 μm, while the anode-to-gate distance is set to 5 mm. Long-term emission stability of the cathode with SiN$_x$-gate was carried out in voltage feedback loop mode. All measurements were performed at room temperature.

## Data availability

Source data are provided with this paper. The main text data generated in this study have been deposited in the Figshare database under accession code https://figshare.com/s/1041c0e743f817ca75a6. The data that support other findings of this study are presented in Supplementary Information and available from the corresponding author on request.

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

## Acknowledgements

P.Y.S. acknowledges the financial support from the National Natural Science Foundation of China (NSFC No. 11927812 and 52202469) and the National Key Research and Development Program of China (No. 2020YFC2201004). L.M.S. acknowledges the financial support from the National Natural Science Foundation of China (NSFC No. 51902112). The authors acknowledge engineer P.L. in the Center of Optoelectronic Micro&Nano Fabrication and Characterizing Facility, Wuhan National Laboratory for Optoelectronics of Huazhong University of Science and Technology for the support in SiN$_x$ films deposition.

## Author contributions

P.Y.S. and L.M.S. conceived the idea. D.Y.X. designed and simulated the cathode based on the SiNx electret. D.Y.X., H.H.D. and Y.R.W. fabricated the cathode. X.C.S. and Y.L.Z. contributed to the simulation analysis for the cathode. D.Y.X., S.L.Z. and H.H.D. conducted the current measurements. S.Y.K. contributed to the surface potential measurements of the gate. F.J.H., L.C.T. and D.R.Y. provided comments on mechanism and data analysis. D.Y.X. drafted the manuscript and all authors provided comments. P.Y.S. and L.M.S. supervised the research.

## Competing interests

The authors declare no competing interests.
