## [Peer Review File · Nature Communications]

Boosting the Electron Beam Transmittance of Field Emission Cathode Using a Self-Charging GateREVIEWER COMMENTS

Reviewer #1 (Remarks to the Author):

Please see the attachment.

Reviewer #2 (Remarks to the Author):

In this manuscript, the authors proposed a self-charging gate by injecting tunneling electrons on the surface of SiNx electret film to enhance electron beam transmittance of field emission cathode. Unfortunately, based on the below issues, the manuscript in current version is not yet publishable in Nature Communication. I recommend this draft to be further concerned after major revision.

1. The high transmittance of 96.17% is achieved under an extremely low current of 27 μA , while it decreases to 91.84% at 101.59 μA . Comparatively, the reference in Figure 6 can obtain 90% in transmittance at a much high current of 10 mA. The most significant highlight of this draft (high transmittance) is greatly depressed.
2. The author stated that charging the cathode for almost 12 h. I suggest the author to further investigate the effect of charging period on electron beam transmittance of field emission cathode, since the electrons move fast in vacuum environment.
3. This work states that an electret layer (SiNx) with a thickness of 1 μm was deposited onto the Au/Si to achieve the electron focusing. However, the thickness of SiNx electret would have influence on the charge injection property. I recommend the author to further investigate the thickness effect.
4. The author should clarify the reason why thermocompression processes technology can weak the shielding effect of field emitters in Page 6 Line 6.
5. The simulation results in Figure 3e and 3f indicate that the localized electric field strength of device using Au/Si gate is quite larger than that of device using SiNx/Au/Si gate, which is not favorable for obtaining a high field emission current density. Please clarify it.
6. It is inconvenient to compare the result with other reported data in Fig. 6. It is suggested that they add the reference No. to the cited data. In addition, there is no obvious physical meaning to compare these results on current. The authors shall make comparison on current density. What is more, the current density of this work is relatively lower ($\sim 0.33 \text{ mA}\cdot\text{cm}^{-2}$) than that of the Ref [92] in the figure.
7. There is a typo in Page 2 Line 5: “..... such as its ultralow turn on voltage ($> 0.4 \text{ V}/\mu\text{m}$)” should be rectified to “ $< 0.4 \text{ V}/\mu\text{m}$ or $\sim 0.4 \text{ V}/\mu\text{m}$ ”.
8. There is a typo in Page 12 Line 19: “transmittance $I_g/(I_g+I_a)$ ” should be rectified to “ $I_a/(I_g+I_a)$ ”.

Reviewer #3 (Remarks to the Author):

The paper presents a novel approach on the Electron Beam Transmittance of Field Emission Cathode Using a Self-Charging Gate. The paper in principle is well written and presents new information on increasing the efficiency of FE cathodes. In spite of this there are point that require attention and need revision:

1. The charge trapping in SiNx is still a topic under investigation. Stoichiometry affects significantly the dielectric film conductivity which increases with increasing silicon content as demonstrated in (DOI: 10.1109/JMEMS.2011.2167670) and (DOI: 10.1109/JMEMS.2019.2962068). Thus the SiNx stoichiometry is very important for those who intent to implement the presented paper and therefore if the authors have any information it should be included in the revised paper.
2. Page 6 line 23 that authors state: " ... (ions or electrons) can be stored in the electret material to form the positive or negative Vs, leading to the charging of opposite polarity in the metal electrode. [65]. ". But the reference [65] refers to organic electret/s where dipoles are present while in SiNx the presence of dipoles are still under consideration. Moreover, electrons are injected into SiNx while positive ions will be attached at the dielectric film surface and this depending on the accelerating electric field intensity. This part needs reconsideration.
3. In page 7, Fig.2c the authors present the PECVD SiNx band structure as being a crystalline material but in fact it is amorphous with band-tails etc. Moreover, depending on the deposition conditions: gas flow, plasma frequency and RF power as well as the substrate temperature (information that must be included) the generated trap distribution across band gap will vary. So, Fig.2c needs to be reconsidered.
4. In page 9 line 26 the authors state: "The increase in surface conductance and water evaporation accelerate the charge decay on the SiNx electret surface.[83]". For the case of SiNx a detailed study of surface charge decay has been presented in "(DOI 10.1088/0957-4484/22/3/035705) which presents the surface charge decay rate vs ambient humidity.
5. The hydrogen "outgassing" and electrons extraction from the SiNx gate must be carefully presented in the supplement.

The manuscript provides a somewhat interesting approach using an electret gate to obtain high-transmittance from the field emission device based on carbon nanotube (CNT) cathode. The introduction of “self-charging” SiN_x/Au/Si gate would be helpful for e-beam focusing by a surface potential, enhancing the transmittance of electron-beams (e-beams). However, there are little physical insights and/or technical breakthroughs in field emission. In general, charging or space charges in an insulator is well known to affect strongly electric fields nearby, and so field emission behaviors. Furthermore, charging of an insulator in electron pathways is strictly prohibited because it inevitably causes instability in e-beam generation and focusing, deteriorating the device performances. Charging is very uncontrollable, prone to changes upon external stimuli like thermal energy, light, electrical agitation, etc. Even though the authors showed the long-term current stability in Fig. 5 in the manuscript, their measurement was performed at a very low current level, ~25uA (current density ~ 331 uA/cm²). They should estimate the stability at even higher currents ~mA and check the effects of external stimuli on stability and reliability of the charging of SiN_x. I recommend that the manuscript would not be published in Nature Comm., instead, it would be published in another special Journal like J. Vac. Sci. & Tech if the followings are addressed.

1. What about the long-term current stability at higher current levels of several mA, and upon the external stimuli?
2. After the negative charging of the electret-gate surface is saturated, what happens to the e-beam trajectory (transmittance) and the gate surface potential if the cathode potential is lowered to increase the emission current?
3. How to measure the I-V characteristics in your “triode” configuration?
According to the mechanism in the manuscript, the electret-gate surface charge saturates after 12 hours, and then the field between the cathode and gate electrode would be changed by amount of the charging.
I wonder if the I-V data was derived after fully considering these effects.
4. What is the relation between the thickness of the SiN_x layer and the amount of the surface charging? And what is the effect of e-beam transmittance by thickness?
5. The transmittance shown in Fig. 4C for the Au-gate is extraordinarily low, blow 30%. There may be an artifact in measuring the transmittance or strong misalignment between the gate hole and CNT emitter, not giving a good reference for comparison of SiN_x-gate and Au-gate.

Responses to Reviewers' Comments (NCOMMS-23-23925A)

Response to Reviewer 1

General comments:

The manuscript provides a somewhat interesting approach using an electret gate to obtain high transmittance from the field emission device based on carbon nanotube (CNT) cathode. The introduction of “self-charging” SiN_x/Au/Si gate would be helpful for e-beam focusing by a surface potential, enhancing the transmittance of electron-beams (e-beams). However, there are little physical insights and/or technical breakthroughs in field emission. In general, charging or space charges in an insulator is well known to affect strongly electric fields nearby, and so field emission behaviors. Furthermore, charging of an insulator in electron pathways is strictly prohibited because it inevitably causes instability in e-beam generation and focusing, deteriorating the device performances. Charging is very uncontrollable, prone to changes upon external stimuli like thermal energy, light, electrical agitation, etc. Even though the authors showed the long-term current stability in Fig. 5 in the manuscript, their measurement was performed at a very low current level, ~25 uA (current density ~ 331 uA/cm²). They should estimate the stability at even higher currents ~mA and check the effects of external stimuli on stability and reliability of the charging of SiN_x. I recommend that the manuscript would not be published in Nature Comm., instead, it would be published in another special Journal like J. Vac. Sci. & Tech if the followings are addressed.

Response to General Comments:

We sincerely appreciate your valuable time spent reviewing our manuscript and your thoughtful, constructive feedback. Your recognition of our work is greatly valued. To comprehensively address your comments and address any concerns, we have made substantial enhancements to our manuscript. These include the incorporation of a significant volume of new data, thorough analyses, extensive discussions, and detailed clarifications (**with revised two Figures, newly added eighteen Supplementary Figures, and six Responded Figures**).

Distinguishing itself from conventional insulators, SiN_x electret exhibits exceptional charge capture and storage capabilities due to the charges can be trapped and restricted by the dangling bonds, resulting in the surface charging. Consequently, it maintains a stable surface potential and ensures reliable electron focusing as the gate. Our investigation demonstrates that the post-charging of SiN_x results in a modest negative surface potential and primarily influences the local electric field distribution near the SiN_x/Au/Si gate, exerting minimal impact on the field emission behavior of the cathode emitters (see Fig. 3c-d). To delve deeper into the reliability of SiN_x gate charging, we conducted extensive assessments under various external stimuli, including mechanical impact, vibration, light, thermal, and electrical energies from the discharging mechanism to the real test. Although external stimuli would contribute to

electron extraction, our experiment results consistently demonstrate that external stimuli do not significantly influence the charging stability of SiN_x as well as the transmittance of the cathode. Lastly, we conducted separate tests to evaluate the long-term current stability of the cathode under external stimuli and a high current (3.25 mA, corresponding to current density of 9.50 mA/cm²) conditions. These assessments confirm the cathode's sustained operation under external stimuli and high current/current density, further affirming the stability and reliability of SiN_x charging.

In the subsequent sections, we will provide point-by-point responses to your comments. Any newly added contents in both the revised manuscript and Supplementary Information (SI) are clearly indicated in **red and highlighted** for your convenience.

Specific Comments:

Comment 1:

What about the long-term current stability at higher current levels of several mA, and upon the external stimuli?

Response to comment 1:

We thank the reviewer for the constructive comments and suggestions. We have performed experiments to verify the long-term current stability at a higher current level of ~3.25 mA (Fig. S17). Moreover, we further analyzed external stimuli that could potentially impact the charging performance of SiN_x and conducted experiments to demonstrate the stability and reliability of the charging of SiN_x upon external stimuli (Fig. S18-S30). Despite external stimuli potentially affecting electron extraction, our experimental results consistently indicate that these stimuli do not substantially impact the charging stability of SiN_x or the transmittance of the cathode. The long-term current stability results of the cathode under external stimuli and high current also demonstrate the stability and reliability of SiN_x charging. We have integrated the following paragraphs into both the revised manuscript and the SI to offer enhanced elucidation on these matters.

[On Page 18 of the revised manuscript and Pages 25-26 of the revised SI (Fig. S17)]

To verify the long-term stability of CNTs field emission cathode (FEC) based on SiN_x-gate under a high current level (~mA), we conducted an enduring stability test on cathode emission at around ~3 mA. As illustrated in Fig. S17a, over a continuous 60 hours field emission test, the cathode maintains an average emission current of 3.25 mA (corresponding to the emitter area of 0.342 cm² and current density of 9.50 mA/cm²), with a current fluctuation of 0.49%. The average E-beam transmittance through the SiN_x-gate is measured at 95.30%, with a transmittance fluctuation of 0.30% (Fig. S17b). The test outcomes demonstrate that even under high current (3.25 mA) and current density (9.50 mA/cm²), the SiN_x-gate can sustain its charging characteristics and focusing effects, enabling the achievement of a long-term stable and high E-beam transmittance for mA-level CNTs FEC.

However, further improvement of current density upon this foundation (9.50

mA/cm²) may exert influences on the cathode's long-term stability. This is because, under higher current density conditions, the instability of the CNTs emitters, such as carbon evaporation and adsorption resulting from thermal effects¹, could potentially affect the performance of the SiN_x-gate, consequently impacting the E-beam transmittance of the cathode. Therefore, to enhance the cathode's emission current range while maintaining a high E-beam transmittance, the focus should be on increasing the emission area of the CNTs rather than the current density, resulting from a larger emitting area can generate a higher current level.

Figure S17. Long-term current measurements of the cathode with SiN_x-gate. (a) Emission current stability of the cathode at 3.25 mA. (b) Transmittance of the cathode.

[On Page 18 of the revised manuscript and Pages 26-37 of the revised SI (Fig. S18-S29)]

The charging performance of SiN_x may be influenced by external stimuli, thereby affecting the performance of the cathode. Therefore, we further analyzed parameters that could potentially impact the charging performance of SiN_x shown in Table S2, such as environmental vibrations, thermal energy, light energy, electrical energy, etc.

Table S2: Parameters affecting the charging performance of SiN_x.

Parameter	Influencing mechanism	Reference
Mechanical force and temperature cycling	Film degradation	2
Thermal energy	Thermal-stimulated discharge	3,4
Light energy	Photo-stimulated discharge	5,6
Electrical energy	Charge distribution	7

- Mechanical force and temperature cycling

For the application of CNTs FEC in space exploration, ground-based environmental simulation tests are essential. In accordance with aerospace standards, before the launch of the TianQin-1 satellite, a comprehensive ground-based environmental performance verification is required to ensure the system's reliability². This verification process consists of two essential components: mechanical performance testing and temperature performance testing. The mechanical performance tests entail impact and vibration tests, while the temperature performance tests encompass the temperature cycle test at atmospheric pressure and thermal vacuum. Hence, we initially investigated the impact of ground-based environmental simulation tests on the performances of SiN_x-gate and cathode.

As depicted in Fig. S18a, to facilitate environmental simulation testing, an engineering prototype of the FEC was designed. The upper and lower covers of the prototype were fabricated using PEEK insulation material, while the cathode signal was routed through the underlying circuit board. Finally, Fig. S18b illustrates the physical representation of the fabricated engineering prototype of the cathode.

Figure S18. The design of the engineering prototype of the cathode. (a) Schematic and (b) physical images of FEC engineering prototype based on CNTs array emitter.

The parameters for the temperature cycle tests at atmospheric pressure and thermal vacuum are presented in Table S3.

Table S3: Parameters of the temperature performance test.

Parameter	Atmospheric pressure thermal cycling	Vacuum thermal cycling
Test pressure	Atmospheric pressure	$<6.65 \times 10^{-3}$ Pa
Test temperature	-25 °C-60 °C	-25 °C-60 °C
Cycle	13.5	3.5
Warming rate	3 °C-5 °C/min	1 °C-3 °C/min

Fig. S19a depicts the experimental instrument (JTK-225L-B) utilized in the ambient pressure thermal cycling experiment (left), concurrently illustrating the

configuration of the cathode within the instrument (right). Fig. 19b illustrates the temperature profile for the atmospheric pressure thermal cycling, it can be observed from the figure that the test curve closely adheres to the specified parameter requirements. Fig. S20a depicts the experimental instrument (KM0812-CPLN) utilized in the vacuum thermal cycling experiment (left), concurrently illustrating the configuration of the cathode within the instrument (right). Fig. S20b illustrates the temperature profile for the vacuum thermal cycling, it can be observed from the figure that the test curve closely adheres to the specified parameter requirements. Following the thermal cycling experiments, no significant changes were observed in the morphology of the cathode.

Figure S19. Atmospheric thermal cycling. (a) The experimental instrument and (b) the test curve for atmospheric thermal cycling of the CNTs FEC, respectively.

Figure S20. Vacuum thermal cycling. (a) The experimental instrument and (b) the test curve for vacuum thermal cycling of the CNTs FEC, respectively.

After completing the temperature performance tests, we conducted impact tests using SY14A-100 (Fig. S21a) to verify the impact resistance of the cathode. The impact test necessitates an impact force of 700 g along each axis. The impact characteristics should exhibit a range of up to 6 dB/octave between 100 Hz and 600 Hz, and up to 700 g between 600 Hz and 4000 Hz, as depicted in Fig. S21b.

Figure S21. The impact tests. (a) The experimental instrument and (b) the test curves for impact tests of the CNTs FEC, respectively.

In the vibration tests, sinusoidal and random vibrations were applied along the three orthogonal axes using ES-10-240 (Fig. S22). The sinusoidal vibration has an amplitude of 10 g within the frequency range of 20 Hz to 100 Hz, with a scan rate of 4 octaves/min shown in Fig. S23. The total root mean square of the random vibration is 9.64 g, and the load time is 1 min. And the power spectral density of the random vibration is depicted in Fig. S24. Following the test, we conducted a verification of the electrical connection and surface topography of the cathode prototype. The results confirm that the cathode prototype successfully passed the mechanical performance tests.

Figure S22. The experimental instrument for vibration tests of the CNTs FEC.

Figure S23. The sinusoidal vibration spectra along the (a) X, (b) Y, and (c) Z axis of CNTs FEC.

Figure S24. The random vibration spectra along the (a) X, (b) Y, and (c) Z axis of CNTs FEC.

After conducting the ground-based environmental simulation tests, a comparison was made between the pre-and post-test E-beam transmittance of the cathode. The results clearly demonstrate that there are no significant changes in the transmittance of the cathode (Fig. S25). This further supports the robustness and reliability of the cathode and gate under external stimuli.

Figure S25. The transmittances versus cathode current plots before and after the environmental simulation.

● Thermal energy

After reaching saturation upon charging, the internally stored space charges within electret materials are typically in a frozen state⁸. However, when electrets are subjected to heating, the mobility of internal space charges increase rapidly. Consequently, thermal stimulation significantly reduces the decay time of the electret's charges. The positions and shapes of peaks in the current-temperature spectrum generated by thermal-stimulated discharge reflect effectively the microscopic characteristics of stored charges within the electret materials. Extensive research into the thermal-stimulated discharge characteristics of SiN_x electret has been carried out in previous works, as depicted in Fig. R1^{3,4}. Thermal stimulation discharge occurs when the SiN_x electret is heated above 300 °C, thereby altering the charge distribution within the electret.

However, in contrast to thermionic cathodes, CNTs-based cathode (named cold cathode) can operate at room temperature^{9, 10}. Therefore, the impact of thermal-stimulated discharge on the charging performance of SiN_x is not considered in practical processes. Moreover, during the cathode's actual operational process, electrons produced by CNTs continuously emit onto the SiN_x-gate surface. Thus, the SiN_x electret should be in a state of real-time charge saturation. In conclusion, thermal stimulation does not affect the high transmittance characteristics of CNTs FEC.

Figure R1. The thermal-stimulated discharge of electrets. (a) Normalized thermal

stimulation discharge curve of a negatively charged sample of Si_3N_4^3 . (b) Thermal stimulation discharge spectra for $\text{Si}_3\text{N}_4/\text{SiO}_2$ electret⁴.

● Light energy

Additionally, similar to thermal-stimulated discharge, photo-stimulation can induce discharge phenomena in electret materials. A. Mellinger et al. have extensively studied photo-stimulated discharge in electret materials^{5, 6}. Under ultraviolet (UV) light irradiation, electrets exhibit a weak discharge phenomenon, affecting the internal charge distribution within the material. Based on this, to investigate the impact of light irradiation on the charging performance of SiN_x electret, we conducted the following experiment illustrated in Fig. S26a. We utilized the setup to measure the current of the SiN_x electret before and after charging under the irradiation of the UV light source (TUD59H1B, Sensor Electronic Technology, Inc) with peak emission wavelengths from 250 nm to 260 nm. From Fig. S26b-c, it can be observed that when the UV light source is modulated at frequencies of 0.1 Hz and 0.5 Hz, the discharged current of the charged SiN_x reaches several tens of pA, whereas the discharged current of the SiN_x electret before charging remains very small. The test results indicate the presence of weak photo-stimulated discharge in the SiN_x electret.

Figure S26. The photo-stimulated discharge of SiN_x electret. (a) The schematic diagram of UV irradiation for the SiN_x electret before and after charging. The photo-stimulated current under (b) 0.1 Hz UV modulation and (c) 0.5 Hz UV modulation.

To investigate the impact of photo-stimulated discharge on the transmittance of CNTs FEC, we employed the setup depicted in Fig. S27a to measure the E-beam transmittances of the cathode under both UV light irradiation and no UV light irradiation conditions. The test results presented in Fig. S27b clearly indicate that UV

irradiation has minimal effect on the E-beam transmittance of the cathode. Across the emission current range of 0-100 μA , the E-beam transmittances of the cathode remain consistently above 96% regardless of whether UV irradiation is present or not.

Figure S27. The E-beam of SiN_x-gate under UV irradiation. (a) The illustration for FEC testing under UV irradiation. (b) The transmittances versus cathode current plots with and without UV irradiation.

Furthermore, we investigated the short-term stability of E-beam transmittances in CNTs FEC under different UV light modulation frequencies. These modulation frequencies are set at 0 Hz, 0.1 Hz, 1 Hz, and 10 Hz. At an emitted cathode current of 10 μA , E-beam transmittances consistently exceed 96% for all UV modulation frequencies, with fluctuations of less than 0.3% (Fig. S28a-d).

Figure S28. The transmittances of the cathode under different UV modulations of (a) 0 Hz, (b) 0.1 Hz, (c) 1 Hz, and (d) 10 Hz when the cathode current is $\sim 10 \mu\text{A}$. The transmittances of the cathode under different UV modulations of (e) 0 Hz, (f) 0.1 Hz, (g) 1 Hz, and (h) 10 Hz when the cathode current is $\sim 100 \mu\text{A}$.

Similarly, when the emitted cathode current is increased to $100 \mu\text{A}$, E-beam transmittances remain above 93% for various UV modulation frequencies, accompanied by transmittance fluctuations of less than 0.4% (Fig. S28e-h). Experimental outcomes further indicate the negligible impact of photo-stimulation on the E-beam transmittance of CNTs FEC.

● **Electrical energy**

In addition to thermal and photo-stimulated discharge, electrical stimulation could also potentially impact the charging performance of SiN_x , and hence we investigated the variations in E-beam transmittance under different current steps and modulation frequencies. As illustrated in Fig. S29a-b, a 0.1 Hz current modulation was used to check the charging stability of the SiN_x eletret under different current steps (corresponding to different voltage steps). Over a series of 10 current modulation cycles ranging from $\sim 10 \mu\text{A}$ to $\sim 50 \mu\text{A}$, the average E-beam transmittance of the SiN_x -gate remains consistently at 94.98%, exhibiting a remarkably low fluctuation of only 1.6%. In a sequence of 10 cycles during which the current undergoes a stepwise increase from $\sim 10 \mu\text{A}$ to $\sim 100 \mu\text{A}$, the mean E-beam transmittance across the SiN_x -gate electrode persists at 95.74%, exhibiting a marginal variability of only 1.5%.

At the modulation frequency of 0.05 Hz, during transitions of the cathode current from $\sim 10 \mu\text{A}$ to $\sim 50 \mu\text{A}$ (Fig. S29c) and from $\sim 10 \mu\text{A}$ to $\sim 100 \mu\text{A}$ (Fig. S29d), the average E-beam transmittances across the SiN_x -gate electrode are measured at 96.89% and 95.74%, with corresponding transmittance fluctuations of 0.70% and 1.4%, respectively. All the test results consistently demonstrate that despite slight fluctuations in the E-beam transmittance of the SiN_x -gate electrode during electrical stimulation, it can be still maintained at relatively high levels of transmittance. Therefore, the influence of electrical stimulation on the charging stability of SiN_x is minimal.

Figure S29. The transmittances of the cathode during transitions of the cathode current from (a) $\sim 10 \mu\text{A}$ to $\sim 50 \mu\text{A}$ and from (b) $\sim 10 \mu\text{A}$ to $\sim 100 \mu\text{A}$ under 0.1 Hz current modulation. The transmittances of the cathode during transitions of the cathode current from (c) $\sim 10 \mu\text{A}$ to $\sim 50 \mu\text{A}$ and from (d) $\sim 10 \mu\text{A}$ to $\sim 100 \mu\text{A}$ under 0.05 Hz current modulation.

[On Page 18 of the revised manuscript and Page 38 of the revised SI (Fig. S30)]

Following the investigation into the influence of external stimulation on cathode performance outlined in Table S2, a long-term test of 310 hours was conducted on the CNTs FEC to verify its enduring stability. As depicted in Fig. S30a, the average emitted current over the course of the cathode's enduring testing is $26.35 \mu\text{A}$, with a current fluctuation of merely 1.9%. The average E-beam transmittance remains at 97.79% with a fluctuation of 0.89% (Fig. S30b). These results collectively highlight the minimal impact of external stimulation on cathode performances, further validating the reliability of our structural design and the stability of SiN_x 's charging performance.

Figure S30. Long-term current measurements of the cathode with SiN_x -gate. (a) Emission current stability of the cathode at $26.35 \mu\text{A}$. (b) Transmittance of the cathode.

Comment 2:

After the negative charging of the electret-gate surface is saturated, what happens to the e-beam trajectory (transmittance) and the gate surface potential if the cathode potential is lowered to increase the emission current.

Response to comment 2:

We thank the Reviewer for the comments. In the revised manuscript, we have discussed the variations of the E-beam trajectory (transmittance) and the SiN_x -gate surface

potential when the cathode potential is lowered to increase the emission current. The testing and simulation results consistently indicate that, following the charging saturation of SiN_x, there is minimal variation in the surface potential of the SiN_x-gate with increasing cathode voltage (Fig. S12). Consequently, the focusing effect of the SiN_x-gate on electrons remains largely unchanged (Fig. S13). Moreover, we measured the E-beam transmittance of the SiN_x-gate after it reached charging saturation. The results indicate that the cathode achieves an average E-beam transmittance of approximately 96.79% in the current range of 2.14 μ A-1.0 mA, with minimal fluctuations (Fig. 4c). We have incorporated the subsequent paragraphs into the revised manuscript and SI to provide further clarification on these aspects.

[On Pages 14-15 of the revised manuscript and Pages 20-21 of the revised SI (Fig. S12-S13)]

Due to the limited trap density of electret materials, after SiN_x electrets reach charging saturation, the surface potential of the SiN_x-gate will cease to change with increasing charging time¹¹. Similarly, upon achieving charging saturation in the SiN_x-gate, even if we increase the negative high voltage on the cathode to increase the cathode emission current, theoretically, the surface potential of the SiN_x-gate will not change. This is because, at this stage, the electrons trapped within the SiN_x have reached saturation, leading to a stable surface potential (Fig. S12a). To further validate this, we first charged the SiN_x electrets material. Once saturation was achieved, we gradually increased the charging voltage and measured the resulting changes in surface potential. Experimental results indicate that after charging saturation, the impact of voltage on the surface potential of SiN_x is minimal (Fig. S12b).

Figure S12. The charging saturation of SiN_x electret. (a) The illustration for charging saturation of SiN_x electret. (b) Variation of the normalized surface potential of SiN_x electret with charging voltage curve after the charging saturation of the SiN_x electret.

Based on the above analysis, it can be observed that after saturation of the charging of the SiN_x-gate, increasing the cathode voltage does not result in any significant change in the surface potential of the gate. Building upon this foundation, while maintaining the surface potential of the SiN_x-gate constant, we gradually increased the cathode emitter potential to investigate changes in the trajectories of emitted electrons from the cathode. As depicted in Fig. S13, it can be observed that with the SiN_x-gate potential unchanged, as the cathode emitter potential increases from -500 V to -700 V, the SiN_x-gate is capable of achieving E-beam focusing, thereby achieving a high E-

beam transmittance. Consequently, following saturation of charging in the SiN_x-gate, increasing the cathode emission current by increasing the cathode emitter potential will not result in significant alterations to the E-beam trajectories from the cathode.

Figure S13. Electrons trajectories at different cathode potentials of (a) -500 V, (b) -600 V, and (c)-700 V when SiN_x-gate potential remains constant.

[On Pages 14-16 of the revised manuscript (Fig. 4c)]

Following the saturation of charging in SiN_x-gate, we conducted measurements of E-beam transmittance through the SiN_x-gate for cathode emission current ranging from 2.14 μA to 1.0 mA (corresponding to the emission current density of 0.037 mA/cm² to 17.54 mA/cm²). As depicted in Fig. 4c, the test results indicate that during the process of increasing the emission current by enhancing the cathode voltage, the E-beam transmittance through the SiN_x-gate remains insignificantly altered. The average E-beam transmittance across the gate from 2.14 μA to 1.0 mA is measured at 96.79%.

Figure 4c. Transmittance versus cathode current plot.

Both the conducted tests and simulations affirm that throughout the process of increasing the emission current by enhancing the cathode emitter potential, there is no substantial change in the surface potential of the SiN_x-gate, consequently preserving the high E-beam transmittance through the SiN_x-gate.

Comment 3:

How to measure the I - V characteristics in your “triode” configuration? According to the mechanism in the manuscript, the electret-gate surface charge saturates after 12 hours, and then the field between the cathode and gate electrode would be changed by amount of the charging. I wonder if the I - V data was derived after fully considering these effects.

Response to comment 3:

We thank the Reviewer for the comments. We have provided a comprehensive description of the structural setup and methodology employed for measuring the I - V characteristics (Fig. R2). Building upon this foundation, we further conducted measurements of the charging time for the SiN_x-gate and the results highlight that, despite variations in the final emission currents achieved by the cathode, the time for the gate E-beam transmittance to surpass 90% consistently remains under 1 s. Additionally, the E-beam transmittance through the gate reaches its peak value in less than 1.5 s (Fig. S11). Importantly, all the I - V data were acquired after the completion of the SiN_x-gate charging process.

Fig. R2a depicts the schematic diagram of our experimental setup for measuring the I - V curves. We utilized an electrostatic voltmeter to measure the currents on the three electrodes, enabling the characterization of the cathode's I - V curve. The E-beam transmittance of the cathode is calculated according to the current results. Despite measuring the gate's surface potential after 12 hours, the actual field emission process involves the rapid movement of electrons, resulting in prompt charging of the gate surface and subsequent electrons' focusing effects. Fig. R2b demonstrates that within the initial 0.9 seconds of cathode initiation, the gate's E-beam transmittance surpasses 95%, indicating saturation of gate charging. Thus, the I - V curve of the cathode was measured under the charging saturation of the SiN_x-gate. Additionally, the surface charging of the gate predominantly influences the local electric field around the gate apertures and exhibits minimal impact on the CNTs emitter surface. Consequently, the measured I - V curve of the cathode comprehensively accounts for all external influences.

Figure R2. The current measurement of the cathode. (a) The measurement schematic of the triode structure. (b) The emission currents as functions of the turn-on time of the

cathode.

We have incorporated the subsequent paragraphs into both the revised manuscript and the SI to provide a more comprehensive explanation of these issues.

[On Pages 13-14 of the revised manuscript (Fig. 4a) and Pages 19-20 of the revised SI (Fig. S11)]

To further demonstrate the charging speed of the SiN_x-gate surface, we designed the following experiments to detect the current using a high-sampling-rate (2000 Hz) acquisition system immediately after the cathode's initiation. We conducted measurements using different SiN_x-gate samples to assess the changes in cathode and anode currents during the cathode startup process, ranging from 0 to ~125 μA, ~200 μA, ~300 μA, and ~450 μA (corresponding to Fig. S11a-d, respectively), thereby calculating the variations in cathode transmittance.

The test results indicate that at the moment of cathode startup, the emission current is minimal, resulting in a low and unstable E-beam transmittance through the corresponding SiN_x-gate. As the cathode emission current gradually increases, the charging of the SiN_x-gate saturates over time, leading to a gradual increase in the E-beam transmittance through the gate, eventually reaching a stable state. The measured and fitted curves of E-beam transmittance in the figures demonstrate that although the final emission currents attained by the cathode differ, the time required for the gate E-beam transmittance to exceed 90% is consistently less than 1 s. Furthermore, it takes less than 1.5 s for the E-beam transmittance through the gate to reach its maximum value. This not only demonstrates the SiN_x-gate's exceptional electron trapping performance and remarkably swift charging response time but also indicates that CNTs FEC based on SiN_x-gate can be employed in cathode applications necessitating rapid response time.

Figure S11. The charging period of SiN_x-gate. As the cathode current increases from 0 to (a) ~125 μA, (b) ~200 μA, (c) ~300 μA, and (d) ~450 μA, the emission current and transmittance as functions of the turn-on time of cathode.

Comment 4:

What is the relation between the thickness of the SiN_x layer and the amount of the surface charging? And what is the effect of e-beam transmittance by thickness.

Response to comment 4:

We thank the reviewer for the constructive comments and suggestions. In response to your comments and suggestions, we have incorporated a discussion concerning the relationship between the thickness of the SiN_x layer and the extent of surface charging. Additionally, we have explored the impact of E-beam transmittance in relation to the thickness of the SiN_x layer. The experimental results indicate that as the thickness of SiN_x increases, its surface potential gradually rises after charging. However, when the thickness reaches or exceeds 1 μm, the potential increase becomes less significant (Fig. S15a). Additionally, the E-beam transmittance of the SiN_x-gate also increases with the thickness of SiN_x. When the SiN_x thickness reaches or exceeds 1 μm, the gate transmittance saturates (Fig. S15b). We have included the following paragraphs in both the revised manuscript and the SI to provide a more comprehensive explanation of these topics.

[On Page 16 of the revised manuscript and Pages 22-24 of the revised SI (Fig. S15)]

The surface charging of electret materials is manifested through the increase in their surface potential, and there has been a significant number of research on the relationship between the surface potential of electrets and their thickness. Researchers have also derived the expression for the surface potential of electret materials after charging (S1.3)^{12, 13}. In the expression, V_s represents the surface potential, t is the thickness of the electret, $\hat{\sigma}$ stands for the average charge density, ϵ_0 and ϵ_e represent the relative permittivity of the vacuum and the electret material, respectively. It's evident from the formula that the surface potential of the electret material is positively correlated with its thickness, and the relationship has been confirmed by the research conducted by X. Zou et al¹³.

$$V_s = t \hat{\sigma} / \epsilon_0 \epsilon_e \quad \text{S1.3}$$

However, in practical charging processes, the penetration depth of electrons within the electret material is limited, depending on the initial energy of the electrons. As illustrated in Fig. R3, the findings of Z. Gan et al.'s research indicate that when the energy of an E-beam is less than 1 keV, the electron penetration depth within Si₃N₄ is below 500 nm¹⁴. Similarly, the outcomes of V. Leonov et al.'s research demonstrate that the average charging distance of electrons within Si₃N₄ is less than 935 nm under a specific voltage¹¹.

Figure R3. The relationship between electrons penetration depth in dielectric materials and initial electrons energy¹⁴.

Based on this analysis, the SiN_x electret fabricated in this study also exhibits a maximum electron charging distance (penetration depth) under a specific charging voltage. To further investigate the relationship between surface charging and the thickness of SiN_x electrets, we deposited SiN_x electret layers of varying thicknesses (50 nm, 200 nm, 500 nm, 1000 nm, and 1500 nm) on the Au/Si substrate. After reaching charging saturation under a fixed voltage, we measured the relationship between the surface potential and thickness of the SiN_x electrets. As depicted in Fig. S15a, with increasing thickness, the surface potentials of the SiN_x electrets gradually increase. When the thickness of the SiN_x electret reaches 500 nm, further increases in thickness result in minimal changes in surface potential, indicating that the electron penetration depth within the SiN_x electret has reached its maximum at this point (as shown in the inset in Fig. S15a).

Figure S15. The SiN_x thickness effect on the E-beam transmittance. (a) The relationship between normalized surface potential of SiN_x electret and SiN_x thickness, the inset is the schematic of the electron penetration depth within the SiN_x electret. (b) The correlation between E-beam transmittance and SiN_x thickness.

During the operation of a FEC, the magnitude of the surface potential on the SiN_x -gate will influence the focusing effect on the E-beam and subsequently impact the E-beam transmittance of the cathode. To further investigate the relationship between SiN_x thickness and E-beam transmittance, we fabricated gates with different SiN_x electret

thicknesses, specifically 50 nm, 500 nm, 1000 nm, and 1500 nm, and measured the E-beam transmittance of the cathode in the range of approximately 0-1 mA (corresponding to the current density of 0-17.54 mA/cm²).

The test results, shown in Fig. S15b, reveal that with a thickness of 50 nm, the focusing effect of the E-beam by the SiN_x-gate is limited due to the relatively small surface potential of the electret (Fig. S15a), resulting in an average E-beam transmittance of approximately 44.28%. As the thickness increases to 500 nm, the surface potential of the SiN_x electret becomes sufficiently large (Fig. S15a), implying that the electron charging distance is approaching its maximum value. Consequently, the average E-beam transmittance for the 500 nm SiN_x-gate reaches 95.58%.

With further increases in SiN_x thickness (>500 nm), the average E-beam transmittance of the cathode saturates at >96%. It's important to note that for SiN_x thicknesses below 1000 nm, a slight attenuation in transmittance occurs at high currents (>100 μA) for the FEC. This situation could potentially be attributed to two reasons. Firstly, this could be attributed to the increased initial energy of electrons (due to higher cathode voltage) at high currents, leading to greater electron penetration depth within the SiN_x, and consequently causing a small portion of electron loss¹⁴. Secondly, when using PECVD to deposit thinner SiN_x films on both the surface and sidewalls of the gate, the consistency of the deposited SiN_x thickness on the gate sidewalls could be compromised. This inconsistency might lead to a reduced focusing effect on the gate sidewalls at high currents, which could result in the observed behavior.

Therefore, when the cathode emission current falls within the range of approximately 0-1 mA in this study (the corresponding cathode voltage should be less than -700 V), the thickness of the SiN_x electret should be 1 μm or greater.

Comment 5:

The transmittance shown in Fig. 4C for the Au-gate is extraordinarily low, below 30%. There may be an artifact in measuring the transmittance or strong misalignment between the gate hole and CNT emitter; not giving a good reference for comparison of SiN_x-gate and Au-gate.

Response to comment 5:

We thank the Reviewer for the comments. In the following paragraphs, we have provided a comprehensive discussion regarding the alignment process of the cathode's gate and emitter. Due to the utilization of a high-precision bonding machine and meticulously designed alignment markers during the alignment process, we can assure that the assembly or alignment errors do not adversely affect the measurements of the cathode E-beam transmittance.

Indeed, unlike the Spindt-type extractor that requires precise alignment with the emitter for optimal E-beam transmittance (Fig. R4a), the gate-type extractor designed in this study has smaller dimensions, resembling a planar extractor (Fig. R4b). As a result, in practical applications, the alignment between the gate and the CNTs emitter is generally not a major concern. Furthermore, due to the smaller size of the gate, the relative positioning of the emitter and the gate also has minimal impact on E-beam

transmittance.

Figure R4. The illustration of (a) Spindt-type cathode and (b) Gate-type cathode.

However, to ensure the uniformity of cathode fabrication, high-precision bonding machine and alignment markers are employed during the gate and CNTs emitter assembly process, thus guaranteeing the consistency of the cathode. For the bonding process of the CNTs emitter and gate, the flip-chip bonder (FINEPLACER[®] sigma) with sub-micron placement accuracy was used to ensure the good alignment between the gate hole and CNTs emitter. As shown in Fig. R5a, in the initial step, the mobile lens of the bonder accurately detects the marks present on both the cathode emitter and the gate. Following this, a precise vertical alignment of the cathode emitter and gate marks is achieved. Subsequently, the cathode emitter and gate are subjected to thermal bonding using the high-precision bonder, operated at predetermined pressure and temperature settings. The top views of the cathode emitter and gate, featuring distinct marks designed to ensure precise alignment of the CNTs emitter and gate hole during the bonding process, are presented in Fig. R5b-c. The successful bonding process results in a well-aligned configuration between the gate hole and CNT emitter, as depicted in Fig. R5d.

Figure R5. The bonding process of the cathode. (a) Schematic for the bonding process of the CNTs emitter and gate. Schematic and physical images of (b) cathode emitter

and (c) gate before the alignment bonding. (d) Schematic and physical images of the cathode after the alignment bonding.

As shown in Fig. R6a-d, to achieve high-precision alignment of the cathode emitter and gate, the bonder initially captures photographs of the marks on the cathode emitter (mark1 and mark2) and gate (mark1' and mark2') surfaces, documenting their respective shapes. Subsequently, under software control, the bonder aligns the centers of mark1 with marker1', as well as marker2 with marker2', ensuring optimal precision during the bonding process (Fig. R6e-f). This meticulous procedure guarantees the attainment of a high level of accuracy in the bonding process.

Figure R6. The alignment marks on the (a, c) cathode emitter and (b, d) gate. (e, f) The alignment of the cathode emitter and gate through the alignment marks.

We have included the following paragraph in the revised manuscript.

[On Page 21 of the revised manuscript]

During the bonding procedure of the CNTs array emitter and the gate, we utilized the flip-chip bonder (FINEPLACER[®] sigma), which offers sub-micron placement accuracy, ensuring precise alignment between the gate hole and CNTs array emitter.

We deeply appreciate your insightful comments and valuable suggestions again, which have significantly enhanced the quality of our manuscript. We believe that our revised manuscript, along with our responses, adequately addresses your concerns and comments. Thank you for your time and consideration.

References

1. Williams, L. T., Kumsomboone, V. S., Ready, W. J., Walker, M. L. R. Lifetime and failure mechanisms of an arrayed carbon nanotube field emission cathode. *IEEE Trans. Electron Devices*. **57**, 3163-3168 (2010).
 2. Li, Z., et al. Design and performance test of the spaceborne laser in the Tianqin-1 mission. *Opt. Laser Technol.* **141**, 107155 (2021).
 3. Amjadi, H., Sessler, G. M. Charge storage in APCVD silicon nitride. *IEEE CEIDP Annual Report*. **1**, 64-67 (1997).
 4. Kressmann, R., Sessler, G. M., Gunther, P. Space-charge electrets. *Trans. Dielectr. Electr. Insul.* **3**, 607-623 (1996).
 5. Mellinger, A., Gonzalez, F. C., Gerhard-Multhaupt, R. Photostimulated discharge in electret polymers: An alternative approach for investigating deep traps. *IEEE Trans. Dielectr. Electr. Insul.* **11**, 218-226 (2004).
 6. Mellinger, A., Gonzalez, F. C., Gerhard-Multhaupt, R., Santos, L. F., Faria, R. M. Photostimulated discharge of corona and electron-beam charged electret polymers. *IEEE ISDE Proceedings*. (2002).
 7. Papaioannou, G., Coccetti, F., Plana, R. On the modeling of dielectric charging in RF-MEMS capacitive switches. *IEEE SiRF* 108-111 (2010).
 8. Turnhout, J. V. Thermally stimulated discharge of electrets. *Springer*. (2005).
 9. Kang, J. S., Hong, J. H., Park, K. C. High-performance carbon-nanotube-based cold cathode electron beam with low-thermal-expansion gate electrode. *J. Vac. Sci. Technol. B* **36**, 02C104 (2018).
 10. Yuan, X., et al. A gridded high-compression-ratio carbon nanotube cold cathode electron gun. *IEEE Electron Device Lett.* **36**, 399-401 (2015).
 11. Leonov, V., Hoof, C. V., Goedbloed, M., Schaijk, R. V. Charge injection and storage in single-layer and multilayer inorganic electrets based on SiO₂ and Si₃N₄. *IEEE Trans. Dielectr. Electr. Insul.* **19**, 1253-1260 (2012).
 12. Wu, M. L., Wang, D., Wan, L. J. Directed block copolymer self-assembly implemented via surface-embedded electrets. *Nat. Commun.* **7**, 10752 (2016).
 13. Zou, X., Zhang, J. Study on PECVD SiO₂/Si₃N₄ double-layer electrets with different thicknesses. *Sci. China Technol. Sci.* **54**, 2123-2129 (2011).
 14. Zhenghao, G., Cher Ming, T., Guan, Z. Nondestructive void size determination in copper metallization under passivation. *IEEE Trans. Device Mater. Reliab.* **3**, 69-78 (2003).
-

Response to Reviewer 2

General comments:

In this manuscript, the authors proposed a self-charging gate by injecting tunneling electrons on the surface of SiN_x electret film to enhance electron beam transmittance of FEC. Unfortunately, based on the below issues, the manuscript in current version is not yet publishable in Nature Communication. I recommend this draft to be further concerned after major revision.

Response to General Comments:

We sincerely appreciate your valuable time spent reviewing our manuscript and your thoughtful, constructive feedback. Your recognition of our work is greatly valued. To comprehensively address your comments and address any concerns, we have made substantial enhancements to our manuscript. These include the incorporation of a significant volume of new data, thorough analyses, extensive discussions, and detailed clarifications (with revised three Figures, newly added five Supplementary Figures, revised one Supplementary Figure, and two Responded Figures). In the subsequent sections, we will provide point-by-point responses to your comments. Any newly added contents in both the manuscript and Supplementary Information (SI) are clearly indicated in red and highlighted for your convenience.

Specific Comments:

Comment 1:

The high transmittance of 96.17% is achieved under an extremely low current of 27 μ A, while it decreases to 91.84% at 101.59 μ A. Comparatively, the reference in Figure 6 can obtain 90% in transmittance at a much high current of 10 mA. The most significant highlight of this draft (high transmittance) is greatly depressed.

Response to comment 1:

We thank the Reviewer for the comments. To verify the E-beam transmittance of the cathode at higher current levels, we conducted measurement when the cathode emitted a current of 1 mA. The test results indicate that the average E-beam transmittance of the cathode within the 2.14 μ A-1.0 mA range exceeds 96% (Fig. 4c). The high E-beam transmittance greatly enhances the stability of the device during long-term operation. Although Y. Ahn et al. achieved a transmittance of around 90% at the 10 mA level for their designed cathode, their extraction electrode was based on the Spindt-type, with a larger extraction electrode size that significantly reduces the uniformity of the electric field on the emitter surface, thereby hindering the long-term stability of the cathode¹.

To be more precise, it is indeed more appropriate to compare the E-beam transmittances at different cathode current density levels rather than different cathode currents. Due to variations in the emitting area among different cathodes, comparing the emitted currents between them is not meaningful; instead, current density provides a more meaningful measure of the cathode's current emission level. Therefore, we have modified the Fig. 6 to compare the relationship between current density and

transmittance. We have integrated the following paragraphs into the revised manuscript to offer enhanced elucidation on these matters.

[On Pages 14-16 of the revised manuscript (Fig. 4c)]

To validate the E-beam transmittance of the cathode at higher current levels, following the saturation of charging in SiN_x-gate, we conducted measurements of E-beam transmittance through the SiN_x-gate for cathode emission current ranging from 2.14 μA to 1.0 mA (corresponding to the emission current density of 0.037 mA/cm² to 17.54 mA/cm²). As depicted in Fig. 4c, the test results indicate that during the process of increasing the emission current by enhancing the cathode voltage, the E-beam transmittance through the SiN_x-gate remains insignificantly altered. The average E-beam transmittance across the gate from 2.14 μA to 1.0 mA is measured at 96.79%.

Figure 4c. Transmittance versus cathode current plot.

[On Pages 18-19 of the revised manuscript (Fig. 6)]

As depicted in Fig. 6, most gate-type cathodes based on CNTs exhibit E-beam transmittance levels ranging from 50% to 80%. In contrast, our designed FEC achieves an average E-beam transmittance of 96.79% at the current density levels ranging from 0.037 mA/cm² to 17.54 mA/cm², significantly reducing cathode power consumption and enhancing the stability of the FEC during long-term operation.

Figure 6. Comparison of performances of the proposed cathode according to various gate-type cathodes based on CNTs emitter.

Comment 2:

The author stated that charging the cathode for almost 12 h. I suggest the author to further investigate the effect of charging period on electron beam transmittance of FEC, since the electrons move fast in vacuum environment.

Response to comment 2:

We thank the Reviewer for the comments and apologize for the inaccurate statements in the previous statement. According to your comments and suggestions, we further investigated the effect of the charging period on the E-beam transmittance of FEC. In fact, we only measured the surface potential of the gate after the cathode was charged for 12 hours, not implying that the gate charging process required 12 hours. Hence, we further conducted measurements of the charging period for the SiN_x-gate and the results highlight that, despite variations in the final emission currents achieved by the cathode, the time for the gate E-beam transmittance to surpass 90% consistently remains under 1 s. Additionally, the E-beam transmittance through the gate reaches its peak value in less than 1.5 s (Fig. S11). We have incorporated the subsequent paragraphs into the revised manuscript and SI to provide further clarification on these aspects.

[On Pages 13-14 of the revised manuscript (Fig. 4a) and Pages 19-20 of the revised SI (Fig. S11)]

To further investigate the impact of the charging period on the E-beam transmittance of the FEC, we designed the following experiments to detect the current using a high-sampling-rate (2000 Hz) acquisition system immediately after the cathode's initiation. We conducted measurements using different SiN_x-gate samples to assess the changes in cathode and anode currents during the cathode startup process, ranging from 0 to ~125 μ A, ~200 μ A, ~300 μ A, and ~450 μ A (corresponding to Fig. S11a-d, respectively), thereby calculating the variations in cathode transmittance.

The test results indicate that at the moment of cathode startup, the emission current is minimal, resulting in a low and unstable E-beam transmittance through the corresponding SiN_x-gate. As the cathode emission current gradually increases, the charging of the SiN_x-gate saturates over time, leading to a gradual increase in the E-beam transmittance through the gate, eventually reaching a stable state. The measured and fitted curves of E-beam transmittance in the figures demonstrate that although the final emission currents attained by the cathode differ, the time required for the gate E-beam transmittance to exceed 90% is consistently less than 1 s. Furthermore, it takes less than 1.5 s for the E-beam transmittance through the gate to reach its maximum value. This not only demonstrates the SiN_x gate's exceptional electrons trapping performance and remarkably swift charging response time but also indicates that CNTs FEC based on SiN_x-gate can be employed in cathode applications necessitating rapid response time.

Figure S11. The charging period of SiN_x-gate. As the cathode current increases from 0 to (a) ~125 μA, (b) ~200 μA, (c) ~300 μA, and (d) ~450 μA, the emission current and transmittance as functions of the turn-on time of cathode.

Comment 3:

This work states that an electret layer (SiN_x) with a thickness of 1 μm was deposited onto the Au/Si to achieve the electron focusing. However, the thickness of SiN_x electret would have influence on the charge injection property. I recommend the author to further investigate the thickness effect.

Response to comment 3:

We thank the Reviewer for pointing out this issue. In response to your comments and suggestions, we have incorporated a discussion concerning the relationship between the thickness of the SiN_x layer and the extent of surface charging. Additionally, we have explored the impact of E-beam transmittance in relation to the thickness of the SiN_x layer. The experimental results indicate that as the thickness of SiN_x increases, its surface potential gradually rises after charging. However, when the thickness reaches or exceeds 1 μm, the potential increase becomes less significant (Fig. S15a). Additionally, the E-beam transmittance of the SiN_x-gate also increases with the thickness of SiN_x. When the SiN_x thickness reaches or exceeds 1 μm, the gate transmittance saturates (Fig. S15b). We have included the following paragraphs in both the revised manuscript and the SI to provide a more comprehensive explanation of these topics.

[On Page 16 of the revised manuscript and Pages 22-24 of the revised SI (Fig. S15)]

The surface charging of electret materials is manifested through the increase in their surface potential, and there have been a significant number of researches on the relationship between the surface potential of electrets and their thickness. Researchers have also derived the expression for the surface potential of electret materials after charging (S1.3)^{2, 3}. In the expression, V_s represents the surface potential, t is the thickness of the electret, $\hat{\sigma}$ stands for the average charge density, ε_0 and ε_e represent the relative permittivity of the vacuum and the electret material, respectively. It's evident from the formula that the surface potential of the electret material is positively correlated with its thickness, and the relationship has been confirmed in previous work³.

$$V_s = t \hat{\sigma} / \varepsilon_0 \varepsilon_e \quad \text{S1.3}$$

However, in practical charging processes, the penetration depth of electrons within the electret material is limited, depending on the initial energy of the electrons. As illustrated in Fig. R1, the findings of Z. Gan et al.'s research indicate that when the energy of an E-beam is less than 1 keV, the electron penetration depth within Si₃N₄ is below 500 nm⁴. Similarly, the outcomes of V. Leonov et al.'s research demonstrate that the average charging distance of electrons within Si₃N₄ is less than 935 nm under a specific voltage⁵.

Figure R1. The relationship between electrons penetration depth in dielectric materials and initial electrons energy³.

Based on this analysis, the SiN_x electret fabricated in this study also exhibits a maximum electron charging distance (penetration depth) under a specific charging voltage. To further investigate the relationship between surface charging and the thickness of SiN_x electrets, we deposited SiN_x electret layers of varying thicknesses (50 nm, 200 nm, 500 nm, 1000 nm, and 1500 nm) on the Au/Si substrate. After reaching charging saturation under a fixed voltage, we measured the relationship between the surface potential and thickness of the SiN_x electrets. As depicted in Fig. S15a, with increasing thickness, the surface potentials of the SiN_x electrets gradually increase. When the thickness of the SiN_x electret reaches 500 nm, further increases in thickness

result in minimal changes in surface potential, indicating that the electron penetration depth within the SiN_x electret has reached its maximum at this point (as shown in the inset in Fig. S15a).

Figure S15. The SiN_x thickness effect on the E-beam transmittance. (a) The relationship between normalized surface potential of SiN_x electret and SiN_x thickness, the inset is the schematic of the electron penetration depth within the SiN_x electret. (b) The correlation between E-beam transmittance and SiN_x thickness.

During the operation of a FEC, the magnitude of the surface potential on the SiN_x-gate will influence the focusing effect on the E-beam and subsequently impact the E-beam transmittance of the cathode. To further investigate the relationship between SiN_x thickness and E-beam transmittance, we fabricated gates with different SiN_x electret thicknesses, specifically 50 nm, 500 nm, 1000 nm, and 1500 nm, and measured the E-beam transmittance of the cathode in the range of approximately 0-1 mA (corresponding to the current density of 0-17.54 mA/cm²).

The test results, shown in Fig. S15b, reveal that with a thickness of 50 nm, the focusing effect of the E-beam by the SiN_x-gate is limited due to the relatively small surface potential of the electret (Fig. S15a), resulting in an average E-beam transmittance of approximately 44.28%. As the thickness increases to 500 nm, the surface potential of the SiN_x electret becomes sufficiently large (Fig. S15a), implying that the electron charging distance is approaching its maximum value. Consequently, the average E-beam transmittance for the 500 nm SiN_x-gate reaches 95.58%.

With further increases in SiN_x thickness (>500 nm), the average E-beam transmittance of the cathode saturates at >96%. It's important to note that for SiN_x thicknesses below 1000 nm, a slight attenuation in transmittance occurs at high currents (>100 μA) for the FEC. This situation could potentially be attributed to two reasons. Firstly, this could be attributed to the increased initial energy of electrons (due to higher cathode voltage) at high currents, leading to greater electron penetration depth within the SiN_x, and consequently causing a small portion of electron loss⁴. Secondly, when using PECVD to deposit thinner SiN_x films on both the surface and sidewalls of the gate, the consistency of the deposited SiN_x thickness on the gate sidewalls could be compromised. This inconsistency might lead to a reduced focusing effect on the gate sidewalls at high currents, which could result in the observed behavior. Therefore, when the cathode emission current falls within the range of approximately 0-1 mA in this study (the corresponding cathode voltage should be less than -700 V), the thickness of

the SiN_x electret should be 1 μm or greater.

Comment 4:

The author should clarify the reason why thermocompression processes technology can weak the shielding effect of field emitters in Page 6 Line 6.

Response to comment 4:

We thank the Reviewer for the comments. Indeed, the thermocompression bonding process plays a pivotal role in transferring CNTs emitter arrays from their initial substrate to a metal substrate. The effective approach for mitigating the field emitter shielding effect lies in the design of CNTs patterns (Fig. S3). To convey this more clearly, we have provided an in-depth explanation of the impact of the shielding effect and the methods employed to suppress its influence in our revised manuscript and SI.

[On Page 6 of the revised manuscript and Pages 8-10 of the revised SI (Fig. S3)]

The thermocompression bonding process is a crucial step used to transfer CNTs emitters from their original substrate to a metal substrate. The veritable strategy to suppress the field emitter shielding effect resides in the patterning design of CNTs, wherein the transformation of bulk CNTs configurations into patterned clusters, known as patterned CNTs clusters, is achieved. Due to the presence of edge effects, larger emitters exhibit stronger edge electric fields, resulting in a concentration of current emission primarily at the edge regions, consequently causing a shielding effect on current emission in the central area^{6, 7}. Utilizing the patterning technique to reduce emitter size, to a certain extent, can suppress the impact of the shielding effect, thereby significantly increasing the cathode's current emission density.

Efforts have been made to choose the dimensions of the emitters array and suppress the shielding effect. For instance, a team led by M. Cole at the University of Cambridge has fabricated an arrayed emitter of single CNT and conducted simulation analyses⁸. As shown in Fig. R2a, the electron emission capability becomes stronger as the spacing between the arrays increases from 1 μm to 10 μm. This conclusion has also been further confirmed, as demonstrated in Fig. R2b, where research results indicate a significant change in current values as the array size increases from 70 μm to 1 mm⁹. In summary, the strength of the shielding effect is highly dependent on the spacing between the arrays.

Figure R2. The shielding effect of the CNT array. (a) A representative β -map

illustrating the 1, 2, 6, 8, and 10 μm pitch CNTs array⁸. (b) Current-voltage characteristics of the carbon fiber cathodes for five different separations: 1 mm, 500 μm , 280 μm , 140 μm , and 70 μm ⁹. (c) Theoretical and experimental variation in $\beta_{\text{array}}/\beta_{\text{emitter}}$ as a function of the emitter pitch-to-height ratio⁸.

By modeling and simulating arrayed one-dimensional materials, researchers have equivalently characterized the shielding effect as the ratio of the overall field enhancement to the field enhancement of a single element. A stronger shielding effect corresponds to a lower value of this ratio. The modeling and simulation results demonstrate that the strength of the shielding effect is determined by both the spacing between the arrays and the height of the emitter. This ratio can be expressed using the following equation¹⁰:

$$\frac{\beta}{\beta_0} = 1 - \exp\left(-a\left(\frac{b}{h}\right)^c\right) \quad \text{S1.2}$$

β_0 represents the field emission enhancement factor of a single CNT, β denotes the average field emission enhancement factor of the cathode, a and c are constants, b represents the array pitch, and h represents the emitter height^{11, 12}. As indicated by equation S1.2, the shielding effect is related to the ratio of the emitter array pitch and height. Their relationship is depicted in Fig. R2c⁸. When the value of b/h exceeds 2, the relative field enhancement effect saturates. In other words, when the ratio of array pitch to height exceeds 2, the emission performance of the cathode reaches its optimal state, with the influence of the shielding effect on electron emission capability minimized^{13, 14}. This suggests that increasing the spacing between emitters can reduce the shielding effect, allowing for more uniform emission from multiple emitters and enhancing the overall field enhancement capability.

Based on the above analysis, it is evident that to suppress the impact of the shielding effect and achieve uniform field emission, and it is desirable to have a ratio of the spacing to height of a single CNT greater than 2. However, in practical applications, controlling the pitch of each CNT is challenging. Therefore, we have devised clustered CNTs array to alleviate the shielding effect between emitters. Similar to a single CNT array, clustered CNTs array also exhibits an optimal ratio of pitch to height that minimizes the shielding effect. Thus, we have designed CNTs clusters with different cluster sizes of 240 μm , 480 μm , and 720 μm , and employed CST simulation to investigate the variation of cathode current density with changes in the ratio of cluster pitch to height under the same voltage. As depicted in Fig. S3a, when the ratio of CNTs array pitch to height exceeds 5, the current density reaches saturation, indicating the minimal screening effect between the CNTs array.

By employing the described thermocompression process, while taking into account the processing limitations, we fabricated three cathodes with different patterning sizes of emitters (240 μm , 480 μm , and 720 μm) and conducted field emission testing. The ratio of array spacing to height was designed to be 6. From the test results shown in Fig. S3b, it can be observed that the maximum current densities for the CNTs arrays with a side length of 240 μm , 480 μm , and 720 μm are 13.72

mA/cm^2 , $5.97 \text{ mA}/\text{cm}^2$, and $3.34 \text{ mA}/\text{cm}^2$, respectively. This indicates that, under the same cathode voltage, the $240 \mu\text{m}$ unit possesses stronger electron emission capability, exhibiting the most significant suppression of the shielding effect. Therefore, all subsequent experimental designs were based on the $240 \mu\text{m}$ CNTs array.

Figure S3. The shielding effect of the cathode emitter. (a) Simulation for the variation of current density (J) as a function of the emitter pitch-to-height ratio, with three different side lengths for CNTs clusters: $240 \mu\text{m}$, $480 \mu\text{m}$, and $720 \mu\text{m}$. (b) The measured current density for the CNTs array with a side length of $240 \mu\text{m}$, $480 \mu\text{m}$, and $720 \mu\text{m}$ under the same emitter pitch-to-height ratio.

Comment 5:

The simulation results in Figure 3e and 3f indicate that the localized electric field strength of device using Au/Si gate is quite larger than that of device using $\text{SiN}_x/\text{Au}/\text{Si}$ gate, which is not favorable for obtaining a high field emission current density. Please clarify it.

Response to comment 5:

We thank the Reviewer for pointing out this issue. In our efforts to address your comments, we have first made corrections to the electric field simulation results in Fig. 3d. Additionally, we have conducted a comprehensive analysis of the extent to which electric field strength affects the cathode's emission current, and results indicate the impact of the SiN_x -gate on the emitter's electric field is not significant (Fig. S9). Furthermore, we have provided a potential strategy for enhancing the cathode's emission current (Fig. S10), which can significantly enhance the current density of the cathode emitters, thereby substantially reducing the impact of the SiN_x -gate on the emitter's electric field strength.

Due to the negative surface potential of the SiN_x -gate during the operation of the FEC, the electric field strength between the gate and the emitter is weakened to some extent. The surface potential after charging saturation of the SiN_x -gate is only about -60V . Therefore, in practical processes, the suppression of the electric field strength on the cathode emitter due to the lower SiN_x surface potential is relatively minor. To provide a more accurate comparison of the electric field distribution in the presence and absence of SiN_x , we adjusted the surface potential of SiN_x in Fig. 3d to -60 V . Based on

this adjustment, we conducted a new electric field distribution simulation for the cathode, the results of which are shown in Fig. 3c-d. From these figures, it can be observed that the impact of the SiN_x-gate on the emitter's electric field is not significant. We have incorporated the subsequent paragraphs into both the revised manuscript and the SI to offer a more comprehensive explanation of these concepts.

[On Page 11 of the revised manuscript (Fig. 3c-d)]

Figure 3c-d. The potential and electric field distributions between the CNTs array and the anode in the cathode equipped with (c) Au-gate and (d) SiN_x-gate, respectively.

[On Page 12 of the revised manuscript and Pages 17-18 of the revised SI (Fig. S9-S10)]

To further quantitatively describe the surface electric field strength of the cathode emitter with and without the SiN_x-gate, we established a three-dimensional simulation model as depicted in Fig. S9a. Subsequently, we separately simulated the electric field strength on the surface of the CNTs emitter, as shown in Fig. S9b. Without SiN_x, the average electric field strength on the CNTs emitter surface is approximately 2.35×10^6 V/m, whereas with SiN_x, the average electric field strength on the CNT emitter surface is approximately 2.07×10^6 V/m. Consequently, the addition of SiN_x results in an 11.91% reduction in the surface electric field strength of the cathode, as compared to the case without SiN_x.

Figure S9. Simulations of the cathode with and without SiN_x electret. (a) Schematic of the simulation models of the cathode. (b) The electric field distributions of CNTs surface with and without SiN_x.

The slight reduction in the surface electric field strength of the CNTs emitter does indeed contribute to a decrease in the cathode's current emission density to some extent. Hence, to further enhance the current density of the FEC, we performed interface modification on the emitter material. The modified CNTs emitters exhibited significantly increased current density. As shown in the test results of Fig. S10, the maximum current density of the modified cathode is 17.54 mA/cm², compared to 1.27 mA/cm² before modification, representing an approximately 13-fold increase in current density. Even at high current densities, the average E-beam transmittance of the cathode's gate remains above 96%, providing further evidence of the feasibility of our cathode design at high current density operation.

Figure S10. The measured current densities of proposed FEC before and after modification of CNTs array emitter.

Comment 6:

It is inconvenient to compare the result with other reported data in Fig. 6. It is suggested that they add the reference No. to the cited data. In addition, there is no obvious physical meaning to compare these results on current. The authors shall make comparison on current density. What is more, the current density of this work is relatively lower (~0.33 mA·cm⁻²) than that of the Ref [92] in the figure.

Response to comment 6:

We thank the reviewer for the constructive comments. Because the emitting area varies among different cathodes, comparing emitted currents between them lacks significance. Instead, current density offers a more meaningful metric for assessing the cathode's current emission level. Firstly, we have modified Fig. 6 to display a comparison between the maximum current density and transmittance results. Additionally, to confirm the E-beam transmittance of the cathode at higher current levels, we conducted measurements when the cathode emitted a current of 1 mA. The test results indicate that the average E-beam transmittance of the cathode within the 2.14 μA-1 mA (corresponding to the current density of 0.037 mA/cm²-17.54 mA/cm²) range exceeds 96% (Fig. 4c). Furthermore, we have performed long-term test (60 hours) on the cathode at the high current density level (9.50 mA/cm²) to further validate the stability of the SiN_x (Fig. S17).

To provide clearer information, we have added reference numbers in Fig. 6 to indicate the cited literature. Additionally, we have made modifications to Fig. 6 to present the comparison results between the maximum current density and transmittance. We have integrated the following paragraphs into both the revised manuscript and the SI to provide a more comprehensive explanation of these concepts.

[On Page 19 of the revised manuscript (Fig. 6)]

Figure 6. Comparison of performances of the proposed cathode according to various gate-type cathodes based on CNTs emitter.

[On Pages 14-16 of the revised manuscript (Fig. 4c)]

To further validate the E-beam transmittance of our designed sample within the higher current levels, we performed additional tests on the E-beam transmittance at the 0-1 mA levels. As depicted in Fig. 4c, the average E-beam transmittance remains above 96% at 2.14 μ A-1 mA ranges (corresponding to the current density levels of 0.037 mA/cm²-17.54 mA/cm²).

Figure 4c. Transmittance versus cathode current plot.

[On Page 18 of the revised manuscript and Pages 25-26 of the revised SI (Fig. S17)]

To verify the long-term stability of CNTs FEC based on SiN_x-gate under a high current level (~mA), we conducted an enduring stability test on cathode emission at around ~3 mA. As illustrated in Fig. S17a, over a continuous 60 hours field emission test, the cathode maintains an average emission current of 3.25 mA (corresponding to the emitter area of 0.342 cm² and current density of 9.50 mA/cm²), with a current fluctuation of 0.49%. The average E-beam transmittance through the SiN_x-gate is measured at 95.30%, with a transmittance fluctuation of 0.30% (Fig. S17b). The test outcomes demonstrate that even under high current (3.25 mA) and current density (9.50 mA/cm²), the SiN_x-gate can sustain its charging characteristics and focusing effects, enabling the achievement of a long-term stable and high E-beam transmittance for mA-level CNTs FEC.

However, further improvement of current density upon this foundation (9.50 mA/cm²) may exert influences on the cathode's long-term stability. This is because, under higher current density conditions, the instability of the CNTs emitters, such as carbon evaporation and adsorption resulting from thermal effects¹⁵, could potentially affect the performance of the SiN_x-gate, consequently impacting the electrons transmittance of the cathode. Therefore, to enhance the cathode's emission current range while maintaining a high E-beam transmittance, the focus should be on increasing the emission area of the CNTs rather than the current density, resulting from a larger emitting area can generate a higher current level.

Figure S17. Long-term current measurements of the cathode with SiN_x-gate. (a) Emission current stability of the cathode at 3.25 mA. (b) Transmittance of the cathode.

Comment 7:

There is a typo in Page 2 Line 5: “..... such as its ultralow turn on voltage ($> 0.4 V/\mu\text{m}$)” should be rectified to “ $< 0.4 V/\mu\text{m}$ or $\sim 0.4 V/\mu\text{m}$ ”.

Response to comment 7:

Thank you for pointing it out. We apologize for our negligence and have made the necessary corrections on Page 2 Line 14 of the revised manuscript.

[On Page 2 Line 14 of the revised manuscript]

“ $\sim 0.4 V/\mu\text{m}$ ”

Comment 8:

There is a typo in Page 12 Line 19: “transmittance $I_g/(I_g+I_a)$ ” should be rectified to “ $I_a/(I_g+I_a)$ ”.

Response to comment 8:

We thank the Reviewer for pointing out this issue. We sincerely apologize for our oversight and have made the necessary corrections on Page 13 Line 1 of the revised manuscript to rectify the issue.

[On Page 13 Line 1 of the revised manuscript]

“ $I_a/(I_g+I_a)$ ”

We deeply appreciate your insightful comments and valuable suggestions again, which have significantly enhanced the quality of our manuscript. We believe that our revised manuscript, along with our responses, adequately addresses your concerns and comments. Thank you for your time and consideration.

References

1. Ahn, Y., et al. Overall control of field emission from carbon nanotube paste-emitters through macro-geometries for high-performance electron source applications. *Carbon*. **189**, 519-529 (2022).
2. Wu, M. L., Wang, D., Wan, L. J. Directed block copolymer self-assembly implemented via surface-embedded electrets. *Nat. Commun.* **7**, 10752 (2016).
3. Zou, X., Zhang, J. Study on PECVD $\text{SiO}_2/\text{Si}_3\text{N}_4$ double-layer electrets with different thicknesses. *Sci. China Technol. Sci.* **54**, 2123-2129 (2011).
4. Zhenghao, G., Cher Ming, T., Guan, Z. Nondestructive void size determination in copper metallization under passivation. *IEEE Trans. Device Mater. Reliab.* **3**, 69-78 (2003).
5. Leonov, V., Hoof, C. V., Goedbloed, M., Schaijk, R. V. Charge injection and storage in single-layer and multilayer inorganic electrets based on SiO_2 and Si_3N_4 . *IEEE Trans. Dielectr. Electr. Insul.* **19**, 1253-1260 (2012).
6. Fujii, S., et al. Efficient field emission from an individual aligned carbon nanotube bundle enhanced by edge effect. *Appl. Phys. Lett.* **90**, 153108 (2007).

-
7. Shiffler, D., et al. Emission uniformity and emission area of explosive field emission cathodes. *Appl. Phys. Lett.* **79**, 2871-2873 (2001).
 8. Cole, M. T., Teo, K. B., Groening, O., Gangloff, L., Legagneux, P., Milne, W. I. Deterministic cold cathode electron emission from carbon nanofibre arrays. *Sci. Rep.* **4**, 1-5 (2014).
 9. Tang, W., Shiffler, D., Golby, K., LaCour, M., Knowles, T. Experimental study of electric field screening by the proximity of two carbon fiber cathodes. *J. Vac. Sci. Technol.* **30**, 061803 (2012).
 10. Harris, J. R., Jensen, K. L., Tang, W., Shiffler, D. A. Control of bulk and edge screening effects in two-dimensional arrays of ungated field emitters. *J. Vac. Sci. Technol. B* **34**, 041215 (2016).
 11. Rezinikina, M. Mathematical modelling of the electric field of carbon nanotube arrays used in cold cathode electron emission devices. *J. Electrostat.* **109**, 103544 (2021).
 12. Harris, J. R., Jensen, K. L., Shiffler, D. A. Dependence of optimal spacing on applied field in ungated field emitter arrays. *AIP Adv.* **5**, 087182 (2015).
 13. Harris, J. R., Jensen, K. L., Shiffler, D. A., Petillo, J. J. Shielding in ungated field emitter arrays. *Appl. Phys. Lett.* **106**, 201603 (2015).
 14. Harris, J. R., Jensen, K. L., Shiffler, D. A. Modelling field emitter arrays using line charge distributions. *J. Phys. D: Appl. Phys.* **48**, 385203 (2015).
 15. Williams, L. T., Kumsomboone, V. S., Ready, W. J., Walker, M. L. R. Lifetime and failure mechanisms of an arrayed carbon nanotube field emission cathode. *IEEE Trans. Electron Devices.* **57**, 3163-3168 (2010).
-

Response to Reviewer 3

General comments:

The paper presents a novel approach on the Electron Beam Transmittance of FEC Using a Self-Charging Gate. The paper in principle is well written and presents new information on increasing the efficiency of FE cathodes. In spite of this there are point that require attention and need revision:

Response to General Comments:

We sincerely appreciate your valuable time spent reviewing our manuscript and your thoughtful, constructive feedback. Your recognition of our work is greatly valued. To comprehensively address your comments and address any concerns, we have made substantial enhancements to our manuscript. These include the incorporation of a significant volume of new data, thorough analyses, extensive discussions, and detailed clarifications (**with revised one Figure, newly added nine Supplementary Figures, and five Responded Figures**). In the subsequent sections, we will provide point-by-point responses to your comments. Any newly added contents in both the manuscript and Supplementary Information (SI) are clearly indicated in **red and highlighted** for your convenience.

Specific Comments:

Comment 1:

The charge trapping in SiN_x is still a topic under investigation. Stoichiometry affects significantly the dielectric film conductivity which increases with increasing silicon content as demonstrated in (DOI: 10.1109/JMEMS.2011.2167670) and (DOI: 10.1109/JMEMS.2019.2962068). Thus the SiN_x stoichiometry is very important for those who intent to implement the presented paper and therefore if the authors have any information it should be included in the revised paper.

Response to comment 1:

We thank the Reviewer for the comments. In response to your comments and suggestions, we have incorporated a discussion concerning the SiN_x stoichiometry and cited the references above mentioned in the revised manuscript appropriately. As a result of different deposition conditions, the performances of SiN_x films deposited under various conditions varies. Hence, the investigation of the stoichiometry of SiN_x becomes quite significant in this context. To gain a more profound insight into the influence of PECVD SiN_x film deposition parameters on material stoichiometry and the charging process, we have introduced a range of deposition parameters of the SiN_x in the revised manuscript (Table S1). These modifications encompass variations in the ratio of reactive gases, RF power, and substrate temperature. By scrutinizing the FT-IR characterization results, we have ultimately established the SiN_x film deposition conditions for this study (Fig. S2). We have included the following paragraphs in both the revised manuscript and the SI to provide a more comprehensive explanation of these topics.

[On Page 6 of the revised manuscript and Pages 4-8 of the revised SI (Fig. S2)]

It is important to note that during the utilization of PECVD for depositing SiN_x films, deposition conditions such as gas flow rates, radio frequency (RF) power, substrate temperature, and others, can influence the material stoichiometry, consequently affecting the charging performance of the SiN_x electrets. Extensive research has investigated the relationship among dielectric charging, material stoichiometry, and deposition parameters in PECVD SiN_x films^{1,2}.

To gain a deeper understanding of how the deposition parameters of PECVD SiN_x films impact both material stoichiometry and the charging process. Various deposition parameters were employed in this study, involving alterations in reactive gas ratio, RF power, and substrate temperature. The samples under investigation comprise PECVD SiN_x electrets with a thickness of 200 nm, deposited onto Si substrate with Au/Ti (200 nm/40 nm) layer through the PECVD (Oxford PlasmaPro 800 Stratum) process. Throughout the deposition process, reactive species such as silane (SiH₄) and ammonia (NH₃) were employed, with nitrogen (N₂) serving as the dilution gas. By employing alternating deposition using high frequency (HF) and low frequency (LF), the compressive stress generated at LF and the tensile stress generated at HF were effectively balanced, resulting in the fabrication of SiN_x films with reduced compressive stress. In this process, the HF and LF are set at 13.5 MHz and 697 kHz, respectively, with corresponding deposition times of 13 s and 7 s per cycle, and the chamber pressure is set at 1000 mTorr. The primary reaction in the deposition process is illustrated as follows:

As described above, SiN_x films were deposited by altering various deposition parameters individually, including the gas ratio (SiH₄/NH₃), RF power, and substrate temperature. The investigated process parameters encompass the following ranges: 1) gas ratio (SiH₄/NH₃) ranging from 0.45 to 1.36; 2) RF power varying from 40 W to 50 W; and 3) substrate temperatures ranging from 100 °C to 300 °C. All SiN_x films were deposited while maintaining constant chamber pressure and RF frequencies. Table S1 presents a compilation of the diverse deposition parameters utilized in this study.

Table S1: Deposition parameters for SiN_x electrets.

SiH ₄ (sccm)	NH ₃ (sccm)	Total gas flow (sccm)	SiH ₄ /NH ₃ gas ratio, r	RF power (W)	Substrate temperature (°C)
15	11	811	1.36	50	100
15	11	811	1.36	45	100
15	11	811	1.36	40	100
15	11	811	1.36	50	300
15	11	811	1.36	45	300
15	11	811	1.36	40	300
10	11	811	0.91	50	100
5	11	811	0.45	50	100
10	11	811	0.91	50	300

After the completion of samples fabrication, the Fourier transform infrared spectroscopy (FT-IR) material characterization technique was employed to investigate the material stoichiometry of various SiN_x films. FT-IR spectroscopy (Nicolet iS50R, Thermo Scientific, Inc.) was conducted to offer insights into the chemical bonds within the dielectric film and their alterations^{1,3}. IR spectra were acquired using a spectrometer in reflection mode at an angle of 70° , spanning the range of 400 cm^{-1} to 4000 cm^{-1} .

Fig. S2a depicts the FT-IR spectra of distinct SiN_x films, deposited under varied RF power conditions, while maintaining a constant substrate temperature ($100\text{ }^\circ\text{C}$) and SiH_4/NH_3 gas ratio ($r = 1.36$). Typically, three categories of bonds, namely Si-N, N-H, and Si-H, are observed. The peaks observed around $1031\text{-}1034\text{ cm}^{-1}$ are attributed to the presence of the Si-N bond¹. The peak at 2156 cm^{-1} corresponds to the Si-H bond and peaks at 1188 cm^{-1} and 3351 cm^{-1} correspond to the N-H bond, indicating the incorporation of hydrogen into the film during growth from the source gases SiH_4 and NH_3 . Fig. S2b demonstrates the FT-IR spectra of distinct SiN_x films, deposited under varied RF power conditions, while maintaining a constant substrate temperature ($300\text{ }^\circ\text{C}$) and SiH_4/NH_3 gas ratio ($r = 1.36$). The peaks of the Si-N bond are observed around $1061\text{-}1064\text{ cm}^{-1}$. The peak at 2171 cm^{-1} corresponds to the Si-H bond and peaks at 1182 cm^{-1} and 3345 cm^{-1} correspond to the N-H bond. The slight peaks position shift of the three bonds could potentially be attributed to variations in power during the SiN_x film deposition process.

Figure S2. The material stoichiometry of various SiN_x films. (a, b) FT-IR spectra of SiN_x films fabricated under varying RF powers. (c) The absorbances of SiN_x film versus RF power within different substrate temperatures. (d, e) FT-IR spectra of SiN_x films fabricated under different gas ratios. (f) The absorbances of SiN_x film versus gas ratio within different substrate temperatures.

In accordance with the analysis presented in Fig. 2b, the surface charging in SiN_x electret is primarily attributed to amphoteric traps induced by dangling bonds, as elucidated by various studies. This model involves the charging of both electron and hole states, attributed to the presence of trivalent Si^{3+} centers. Hence, our focus lies on the content of Si-N bond within the SiN_x films, as it is positively correlated with Si^{3+}

centers. The absorptances of Si–N bond under various deposition temperatures, as extracted from FT-IR analysis, are depicted in Fig. S2c. From the graph, it is evident that with the increase in RF power, there is an upward trend in the absorption peak of Si–N bond. Consequently, higher RF power is more conducive to the formation of Si³⁺ centers, thereby enhancing the charging performance of SiN_x electrets.

Fig. S2d-e illustrate the FT-IR spectra of different SiN_x films, deposited at varying SiH₄/NH₃ gas ratios, while maintaining a constant RF power (50 W), at the substrate temperature of 100 °C and 300 °C, respectively. The absorptances of the Si–N bond under different deposition temperatures, as extracted from the FT-IR analysis, are illustrated in Fig. S2f. Likewise, it's apparent that an increase in gas ratio corresponds to a rising trend in the absorption peak of the Si–N bond¹. Consequently, a higher gas ratio promotes the formation of Si³⁺ centers, thereby enhancing the charging performance of SiN_x electrets. This phenomenon can be attributed to the increase in silicon content within the investigated SiN_x films as the gas ratio increases. Furthermore, the FT-IR results from both Fig. S2c and Fig. S2f consistently indicate higher absorption peaks of the Si–N bond for films grown at lower temperature. This phenomenon can be attributed to the fact that low-temperature PECVD is more conducive to the formation of traps and dangling bonds within the film.

In conclusion, based on the findings presented above, to enhance the charging performance of the electret in this study, the growth conditions for SiN_x are recommended to be set at a gas ratio of 1.36, an RF power of 50 W, and a substrate temperature of 100 °C.

Comment 2:

Page 6 line 23 that authors state: " ... (ions or electrons) can be stored in the electret material to form the positive or negative Vs, leading to the charging of opposite polarity in the metal electrode. [65].". But the reference [65] refers to organic electrets where dipoles are present while in SiN_x the presence of dipoles are still under consideration. Moreover, electrons are injected into SiN_x while positive ions will be attached at the dielectric film surface and this depending on the accelerating electric field intensity. This part needs reconsideration.

Response to comment 2:

We thank the reviewer for the constructive comments and suggestions. We have engaged in a detailed discussion regarding the types of charges present within the SiN_x after charging and the impact of positive ions on the charging of SiN_x. More specifically, due to the non-polar nature of SiN_x, the predominant charge within SiN_x electrets are space charges rather than dipole charges (Fig. S5). Furthermore, we conducted a detailed analysis of the impact mechanism of positive ions on the charging of SiN_x, and the results indicate that external positive ions have minimal influence on the performance of SiN_x-gate (Fig. S6). We have integrated the following paragraphs into both the revised manuscript and the SI to offer enhanced elucidation on these matters.

[On Page 7 of the revised manuscript and Pages 11-14 of the revised SI (Fig. S5-S6)]

As mentioned in Reference 4 (corresponding to Reference 65 in the original manuscript), electret materials possess the capability to store surface charges, space charges, and dipole charges (Fig. R1a)⁴. Non-polar materials such as polytetrafluoroethylene (PTFE), fluorinated ethylene propylene polymer (FEP), silicon dioxide, and silicon nitride primarily store space charges, whereas polar materials like polyvinylidene fluoride (PVDF) exhibit predominant orientationally polarized dipole charges in addition to some space charges. Therefore, it is widely recognized that polar materials with polar molecules can generate dipole charges. In contrast, the non-polar SiN_x material fabricated in this study does not exhibit dipole charges.

Figure R1. The charge distributions of electrets. (a) Schematic diagram illustrating the charge distributions in the electret materials⁴. (b) Experimental setup for forming a thin-film electret using monoenergetic E-beam irradiation charging⁴.

In addition, when employing E-beam irradiation for charging the electret materials (Fig. R1b), only space charges are deposited in the materials⁴. This is consistent with the method of gate charging in FECs, which involves E-beam irradiation. Therefore, in the case of the SiN_x electret investigated in this study, the stored charges primarily consist of space charges, and no dipole charges are present. To further illustrate the charge distribution within the SiN_x after charging, Fig. S5 depicts a schematic representation of the charge distributions before and after SiN_x-gate charging. As observed in the figure, after charging by E-beam irradiation, space charges (electrons) are distributed among different trap levels within the SiN_x while compensating charges (positive charges) are induced on the Au electrode.

Figure S5. The charge distributions of SiN_x/Au/Si gate before and after E-beam irradiation charging.

Moreover, during the operation of the cathode, the external positive ions can be

attached at the SiN_x film surface and affect the performance of the electret. The sources of external positive ions can be categorized into two main types: firstly, there is a minimal presence of positive ions in the external environment, which may migrate to the surface of the SiN_x -gate under the influence of external forces. These scarce positive ions on the surface of SiN_x neutralize with the electrons present on the SiN_x surface. However, the electrons emitted by the cathode are instantaneously injected into the SiN_x -gate. Even if some of these electrons on the SiN_x are neutralized by positive ions, they are replenished by subsequent emitted electrons. Therefore, these exceedingly small quantities of positive ions do not impact the surface potential of SiN_x , and consequently, they do not affect the E-beam transmittance of the cathode.

Secondly, FEC frequently serves as the neutralizer in electric propulsion systems, supplying electrons to neutralize positive ions or positively charged droplets, thereby preventing the spacecraft from accumulating charge. When the FEC and thruster work in coordination, their positioning relationship, as shown in Fig. S6a, involves a certain angle. This is to prevent positively charged ions and droplets emitted by the thruster from splashing onto the surface of the cathode due to the electric field acceleration, which could affect the cathode's field emission performances. More specifically, the thruster emits a plume with a small divergence angle and high velocity in our simulations (Fig. S6b), and the research findings from other studies have also confirmed this phenomenon^{5, 6}. Therefore, under the influence of the electric field acceleration, positively charged ions or droplets are unlikely to splash onto the surface of the SiN_x -gate electrode. In other words, this setup does not impact the charging performance of the SiN_x -gate electrode.

Figure S6. The collaboration between cathode and thruster. (a) Positional relationship of thruster and cathode during synergistic operation. (b) Simulation of the trajectories of charged ions and droplets emitted by the thruster.

According to the research conducted by V. Kleshch et al., when the thruster and FEC operate together (as shown in Fig. R2a), the electrons emitted by the cathode are attracted and deviate from their original trajectory towards the positively charged ions emitted by the thruster⁷. This ensures effective neutralization between electrons and ions (as illustrated in Fig. R2b-c). Furthermore, even if some positively charged ions

have a chance to splash onto the area above the cathode, these ions are neutralized by the electrons emitted by the cathode, thus not affecting the high transmittance characteristics of the SiN_x-gate. In summary, positively charged ions under electric field acceleration do not impact the charging performance of the SiN_x-gate.

Figure R2. The collaboration between cathode and thruster. (a) Schematic diagram of an air-breathing EP thruster with a ring-shaped neutralizer⁷. (b) Axial cross-section of the neutralizer and calculated trajectories of electrons (in green) and ions (in magenta)⁷. (c) Zoomed-in view of the cathode design and the computed trajectories of electrons⁷.

Comment 3:

In page 7, Fig.2c the authors present the PECVD SiN_x band structure as being a crystalline material but in fact it is amorphous with band-tails etc. Moreover, depending on the deposition conditions: gas flow, plasma frequency and RF power as well as the substrate temperature (information that must be included) the generated trap distribution across band gap will vary. So, Dif.2c needs to be reconsidered.

Response to comment 3:

We thank the Reviewer for pointing this out and apologize for the inappropriate descriptions in Fig. 2c. We have revised the band structure of SiN_x in Fig. 2c and provided information about the growth conditions of SiN_x. Furthermore, we have extensively discussed the interdependence between the band structures and the growth conditions of SiN_x (Fig. S7). We have incorporated the subsequent paragraphs into the revised manuscript and SI to provide further clarification on these aspects.

[On Pages 7-8 of the revised manuscript (Fig. 2c) and Page 14 of the revised SI]

During the fabrication process of SiN_x, factors such as doping and oxidation can introduce defects and impurities, leading to the formation of defect states within the band structure. Additionally, the highly disordered structure and composition of amorphous SiN_x result in the presence of localized states within the bandgap, known as band-tails⁸. Extensive studies have been conducted on the band-tails of amorphous SiN_x, revealing that these band-tails within the bandgap can form a continuous energy

distribution between the conduction and valence bands^{8,9}.

According to the amphoteric traps theory mentioned in Fig. 2b, electrons are primarily captured by trap levels associated with dangling bonds in SiN_x, leading to the formation of a negative surface potential on the SiN_x surface¹⁰. Fig. 2c illustrates the band structures of SiN_x before and after electrons trapping, showing both defect states (Si³⁺ centers) and band-tails. The defect states can be represented as D⁺, D⁰, and D⁻ states depending on the number of electrons captured by the dangling bonds. When two electrons are captured by the dangling bonds, the band structure of SiN_x, as shown in Fig. 2c, indicates that the trap levels generated by Si³⁺ centers are occupied by electrons, resulting in a negative surface potential of the SiN_x.

Figure 2b-c. (b) Conversion model for the three charges states due to the interaction with electrons and holes. (c) The band structures of SiN_x before and after electrons trapping.

[On Pages 20-21 of the revised manuscript]

During the deposition of the SiN_x film, the flow rates of SiN₄, NH₃, and N₂ were set at 15 sccm, 11 sccm, and 785 sccm, respectively. The HF and LF were set to 13.5 MHz and 697 kHz, with corresponding deposition times of 13 s and 7 s per cycle. The RF power was set to 50 W, and the substrate temperature was maintained at 100 °C.

[On Page 8 of the revised manuscript and Pages 14-15 of the revised SI (Fig. S7)]

As reported by U. Zaghoul et al., gas ratio, RF power, and substrate temperature have been found to influence the trap distributions in SiN_x, thereby affecting the charging and discharging processes¹. The reactive gas ratio has been observed to have a more significant impact on the stoichiometry of SiN_x material compared to the effects of RF power or substrate temperature from the FT-IR results shown in Fig. R3. As the SiH₄/NH₃ gas ratio increases from 0.15 to 0.45, the injected charge density and relaxation time constant gradually decrease. However, for gas ratios of 0.6 and 0.8, the decrease becomes more pronounced. This trend can be attributed to the increased silicon content in the SiN_x films, as confirmed by FT-IR data. The higher silicon content results in a higher leakage current, a greater concentration of defects, and an increased number of charge-trapping centers. Consequently, the charge redistribution paths become more extensive with higher silicon content in the SiN_x film, leading to a larger redistribution current and a smaller charge density at the dielectric surface¹.

Figure R3. FT-IR data comparison for SiN_x films deposited using different (a) gas ratios, (b) RF powers, and (c) substrate temperatures¹.

In summary, the variations in gas ratio, RF power, and substrate temperature can impact the trap level distribution in SiN_x thin films. As shown in Fig. S7, with higher defect concentrations, there are more charge-trapping centers, resulting in multiple charge redistribution paths within the SiN_x material. As a consequence, the charge density at a specific location decreases due to the increased availability of charge pathways within the SiN_x film.

Figure S7. The trap level distributions of SiN_x with different charge-trapping centers.

Comment 4:

*In page 9 line 26 the authors state: “The increase in surface conductance and water evaporation accelerate the charge decay on the SiN_x electret surface.[83]”. For the case of SiN_x a detailed study of surface charge decay has been presented in “(DOI 10.1088/0957-4484/22/3/035705) which presents the surface charge decay rate *cs* ambient humidity.*

Response to comment 4:

We thank the Reviewer for the comments. According to your comment and suggestion, we have appropriately discussed the relationship between surface charge decay and ambient humidity and cited this reference in the revised manuscript. We have incorporated the subsequent paragraphs into both the revised manuscript and the SI to provide a more comprehensive explanation of the issue.

[On Page 10 of the revised manuscript and Pages 16-17 of the revised SI]

U. Zaghoul et al. revealed the correlation between relative humidity and the charging and discharging processes of SiN_x material in both air and nitrogen environments¹¹. As

relative humidity increases, the injected charge density rises while the discharge process accelerates. Several factors contribute to the increase in injected charge density, including the conductivity of the adsorbed water film, the presence of surface charges on the dielectric surface, and the less confined electric field distribution between the tip and sample surface as the adsorbed water layer expands under higher relative humidity levels. Additionally, the relaxation time decreases for both air and nitrogen with increasing humidity. This decrease is primarily attributed to the faster neutralization of surface charges with the external medium and the increased conductivity of the adsorbed water film over the SiN_x surface.

Comment 5:

The hydrogen “outgassing” and electrons extraction from the SiN_x gate must be carefully presented in the supplement.

Response to comment 5:

We totally agree with the Reviewer and appreciate this suggestion. In accordance with your comments and suggestions, we have discussed the hydrogen “outgassing” and electrons extraction from the SiN_x-gate. According to our analysis, the high-temperature annealing of SiN_x films can impact their hydrogen outgassing, thereby affecting the film's structural stress (Fig. R4). However, the proposed field emission cathode operates at room temperature, so the impact of hydrogen outgassing on the SiN_x-gate is minimal. Additionally, the electrons extraction from the SiN_x-gate can be understood as the discharge process of SiN_x. Therefore, we further investigated the discharge mechanism of SiN_x under external stimuli (thermal, light, and electrical stimuli), and all results indicate that while external stimuli may lead to SiN_x discharge, the minimal discharge has little impact on the E-beam transmittance of the SiN_x-gate, thus not affecting the device performances of the cathode (Fig. S26-S30). We have included the following paragraphs in both the revised manuscript and the SI to offer a more comprehensive elucidation of these issues.

[On Page 9 of the revised manuscript and Pages 15-16 of the revised SI]

As shown in Equation S1.1,

in which a certain quantity of hydrogen is incorporated into the SiN_x matrix during the PECVD process. According to the research findings by A. Picciotto et al., the hydrogen content in SiN_x materials can indeed influence the material's stress levels¹². Fig. R4 illustrates the correlation between the annealing temperature of SiN_x and the resulting film stress. The observed effect is likely a result of the outgassing of hydrogen, which is typically present in PECVD SiN_x films and originates from the reaction between SiH₄ and NH₃ within the CVD chamber. This outgassing leads to a reorganization of the nitride structure formed by both HF and LF layers, resulting in a change in stress towards positive values. Furthermore, this outgassing phenomenon becomes more pronounced with increasing annealing temperature. However, in contrast to thermionic

cathodes, CNTs-based FEC can operate at room temperature^{13, 14}. Therefore, the impact of annealing temperature on the stress of SiN_x-gate is not considered in practical processes.

Figure R4. Stress in SiN_x as a function of annealing temperature¹².

Furthermore, based on the elemental composition characterization results of SiN_x presented in Fig. 2d, it is evident that the hydrogen content within the SiN_x electret is extremely low. The primary constituents include Si, N, C, and O. Therefore, in practical applications of SiN_x electret, the impact of hydrogen outgassing on film performances is minimal.

Figure 2d. The EDX spectrum of SiN_x film.

[On Page 18 of the revised manuscript and Pages 33-38 of the revised SI (Fig. S26-S30)]

The charging performance of SiN_x may be influenced by external stimuli, thereby affecting the performance of the cathode. Therefore, we further analyzed parameters that could potentially impact the charging performance of SiN_x shown in Table S2, such as thermal energy, light energy, electrical energy, etc.

Table S2: Parameters affecting the charging performance of SiN_x.

Parameter	Influencing mechanism	Reference
Thermal energy	Thermal-stimulated discharge	15, 16
Light energy	Photo-stimulated discharge	17, 18
Electrical energy	Charge distribution	19

- Thermal energy

After reaching saturation upon charging, the internally stored space charges within electret materials are typically in a frozen state²⁰. However, when electrets are subjected to heating, the mobility of internal space charges increase rapidly. Consequently, thermal stimulation significantly reduces the decay time of the electret's charges. The positions and shapes of peaks in the current-temperature spectrum generated by thermal-stimulated discharge reflect effectively the microscopic characteristics of stored charges within the electret materials. Extensive research into the thermal-stimulated discharge characteristics of SiN_x electret has been carried out in previous works, as depicted in Fig. R5^{15, 16}. Thermal stimulation discharge occurs when the SiN_x electret is heated above 300 °C, thereby altering the charge distribution within the electret.

However, in contrast to thermionic cathodes, CNTs-based cathode (named cold cathode) can operate at room temperature^{13, 14}. Therefore, the impact of thermal-stimulated discharge on the charging performance of SiN_x is not considered in practical processes. Moreover, during the cathode's actual operational process, electrons produced by CNTs continuously emit onto the SiN_x-gate surface. Thus, the SiN_x electret should be in a state of real-time charge saturation. In conclusion, thermal stimulation does not affect the high transmittance characteristics of CNTs FEC.

Figure R5. The thermal-stimulated discharge of electrets. (a) Normalized thermal stimulation discharge curve of a negatively charged sample of Si₃N₄¹⁵. (b) Thermal stimulation discharge spectra for Si₃N₄/SiO₂ electret¹⁶.

- Light energy

Additionally, similar to thermal-stimulated discharge, photo-stimulation can induce discharge phenomena in electret materials. A. Mellinger et al. have extensively studied photo-stimulated discharge in electret materials^{17, 18}. Under ultraviolet (UV) light irradiation, electrets exhibit a weak discharge phenomenon, affecting the internal charge distribution within the material. Based on this, to investigate the impact of light irradiation on the charging performance of SiN_x electret, we conducted the following experiment illustrated in Fig. S26a. We utilized the setup to measure the current of the SiN_x electret before and after charging under the irradiation of the UV light source (TUD59H1B, Sensor Electronic Technology, Inc) with peak emission wavelengths from 250 nm to 260 nm. From Fig. S26b-c, it can be observed that when the UV light

source is modulated at frequencies of 0.1 Hz and 0.5 Hz, the discharged current of the charged SiN_x reaches several tens of pA, whereas the discharged current of the SiN_x electret before charging remains very small. The test results indicate the presence of weak photo-stimulated discharge in the SiN_x electret.

Figure S26. The photo-stimulated discharge of SiN_x electret. (a) The schematic diagram of UV irradiation for the SiN_x electret before and after charging. The photo-stimulated current under (b) 0.1 Hz UV modulation and (c) 0.5 Hz UV modulation.

To investigate the impact of photo-stimulated discharge on the transmittance of CNTs FEC, we employed the setup depicted in Fig. S27a to measure the E-beam transmittances of the cathode under both UV light irradiation and no UV light irradiation conditions. The test results presented in Fig. S27b clearly indicate that UV irradiation has minimal effect on the E-beam transmittance of the cathode. Across the emission current range of 0-100 μA , the E-beam transmittance of the cathode remains consistently above 96% regardless of whether UV irradiation is present or not.

Figure S27. The E-beam of SiN_x -gate under UV irradiation. (a) The illustration for FEC testing under UV irradiation. (b) The transmittances versus cathode current plots with

and without UV irradiation.

Furthermore, we investigated the short-term stability of E-beam transmittances in CNTs FEC under different UV light modulation frequencies. These modulation frequencies are set at 0 Hz, 0.1 Hz, 1 Hz, and 10 Hz. At an emitted cathode current of 10 μA , E-beam transmittances consistently exceed 96% for all UV modulation frequencies, with fluctuations of less than 0.3% (Fig. S28a-d). Similarly, when the emitted cathode current is increased to 100 μA , E-beam transmittances remain above 93% for various UV modulation frequencies, accompanied by transmittance fluctuations of less than 0.4% (Fig. S28e-h). Experimental outcomes further indicate the negligible impact of photo-stimulation on the E-beam transmittance of CNTs FEC.

Figure S28. The transmittances of the cathode under different UV modulations of (a) 0 Hz, (b) 0.1 Hz, (c) 1 Hz, and (d) 10 Hz when the cathode current is $\sim 10 \mu\text{A}$. The transmittances of the cathode under different UV modulation of (e) 0 Hz, (f) 0.1 Hz, (g) 1 Hz, and (h) 10 Hz when the cathode current is $\sim 100 \mu\text{A}$.

● Electrical energy

In addition to thermal and photo-stimulated discharge, electrical stimulation could also potentially impact the charging performance of SiN_x, and hence we investigated the variations in E-beam transmittance under different current steps and modulation frequencies. As illustrated in Fig. S29a-b, a 0.1 Hz current modulation was used to check the charging stability of the SiN_x electret under different current steps (corresponding to different voltage steps). Over a series of 10 current modulation cycles ranging from ~10 μA to ~50 μA, the average E-beam transmittance of the SiN_x-gate remains consistently at 94.98%, exhibiting a remarkably low fluctuation of only 1.6%. In a sequence of 10 cycles during which the current undergoes a stepwise increase from ~10 μA to ~100 μA, the mean E-beam transmittance across the SiN_x-gate electrode persists at 95.74%, exhibiting a marginal variability of only 1.5%.

At the modulation frequency of 0.05 Hz, during transitions of the cathode current from ~10 μA to ~50 μA (Fig. S29c) and from ~10 μA to ~100 μA (Fig. S29d), the average E-beam transmittances across the SiN_x-gate electrode are measured at 96.89% and 95.74%, with corresponding transmittance fluctuations of 0.70% and 1.4%, respectively. All the test results consistently demonstrate that despite slight fluctuations in the E-beam transmittance of the SiN_x-gate electrode during electrical stimulation, it can be still maintained at relatively high levels of transmittance. Therefore, the influence of electrical stimulation on the charging stability of SiN_x is minimal.

Figure S29. The transmittances of the cathode during transitions of the cathode current from (a) ~10 μA to ~50 μA and from (b) ~10 μA to ~100 μA under 0.1 Hz current modulation. The transmittances of the cathode during transitions of the cathode current from (c) ~10 μA to ~50 μA and from (d) ~10 μA to ~100 μA under 0.05 Hz current modulation.

Following the investigation into the influence of external stimulation on cathode performance outlined in Table S2, a long-term test of 310 hours was conducted on the CNTs FEC to verify its enduring stability. As depicted in Fig. S30a, the average emitted

current over the course of the cathode's enduring testing is $26.35 \mu\text{A}$, with a current fluctuation of merely 1.9%. The average E-beam transmittance remains at 97.79% with a fluctuation of 0.89% (Fig. S30b). These results collectively highlight the minimal impact of external stimulation on cathode performances, further validating the reliability of our structural design and the stability of SiN_x 's charging performance.

Figure S30. Long-term current measurements of the cathode with SiN_x -gate. (a) Emission current stability of the cathode at $26.35 \mu\text{A}$. (b) Transmittance of the cathode.

We deeply appreciate your insightful comments and valuable suggestions again, which have significantly enhanced the quality of our manuscript. We believe that our revised manuscript, along with our responses, adequately addresses your concerns and comments. Thank you for your time and consideration.

References

1. Zaghloul, U., et al. Effect of deposition gas ratio, RF power, and substrate temperature on the charging/discharging processes in PECVD silicon nitride films for electrostatic NEMS/MEMS reliability using atomic force microscopy. *JMEMS* **20**, 1395-1418 (2011).
2. Birmpiliotis, D., Stavrinidis, G., Koutsourelis, M., Konstantinidis, G., Papaioannou, G. On the discharge transport mechanisms through the dielectric film in MEMS capacitive switches. *JMEMS* **29**, 202-213 (2020).
3. Malik, P., Gupta, H., Ghosh, S., Srivastava, P. Study of optical properties of single and double layered amorphous silicon nitride films for photovoltaics applications. *Silicon*. **15**, 143-151 (2022).

-
4. Guo, Z., Patil, Y., Shinohara, A., Nagura, K., Yoshida, M., Nakanishi, T. Organic molecular and polymeric electrets toward soft electronics. *Mol. Syst. Des. Eng.* **7**, 537-552 (2022).
 5. Asher, J., Huang, Z., Cui, C., Wang, J. Multi-scale modeling of ionic electrospray emission. *J. Appl. Phys.* **131**, 014902 (2022).
 6. Parmar, S. M., Collins, A. L., Wirz, R. E. Electrospray plume modeling for rapid life and performance analysis. *AIAA SCITECH 2022 Forum.* 1357 (2022).
 7. Kleshch, V. I., Ismagilov, R. R., Mukhin, V. V., Orekhov, A. S., Filatyev, A. S., Obraztsov, A. N. Nano-graphite field-emission cathode for space electric propulsion systems. *Nanotechnol.* **33**, 415201 (2022).
 8. Robertson, J. Electronic structure of silicon nitride. *Philos. Mag. B* **63**, 47-77 (2006).
 9. Robertson, J., Powell, M. J. Gap states in silicon nitride. *Appl. Phys. Lett.* **44**, 415-417 (1984).
 10. Crain, M. M., McNamara, S., Depuy, G., Keynton, R. S. Formation of SiO₂/Si₃N₄/SiO₂ positive and negative electrets on a silicon substrate. *JMEMS.* **25**, 1041-1049 (2016).
 11. Zaghoul, U., Bhushan, B., Pons, P., Papaioannou, G. J., Coccetti, F., Plana, R. On the influence of environment gases, relative humidity and gas purification on dielectric charging/discharging processes in electrostatically driven MEMS/NEMS devices. *Nanotechnol.* **22**, 035705 (2011).
 12. Picciotto, A., Bagolini, A., Bellutti, P., Boscardin, M. Influence of interfaces density and thermal processes on mechanical stress of PECVD silicon nitride. *Appl. Surf. Sci.* **256**, 251-255 (2009).
 13. Kang, J. S., Hong, J. H., Park, K. C. High-performance carbon-nanotube-based cold cathode electron beam with low-thermal-expansion gate electrode. *J. Vac. Sci. Technol. B* **36**, 02C104 (2018).
 14. Yuan, X., et al. A gridded high-compression-ratio carbon nanotube cold cathode electron gun. *IEEE Electron Device Lett.* **36**, 399-401 (2015).
 15. Amjadi, H., Sessler, G. M. Charge storage in APCVD silicon nitride. *IEEE CEIDP Annual Report.* **1**, 64-67 (1997).
 16. Kressmann, R., Sessler, G. M., Gunther, P. Space-charge electrets. *IEEE Trans. Dielectr. Electr. Insul.* **3**, 607-623 (1996).
 17. Mellinger, A., Gonzalez, F. C., Gerhard-Multhaupt, R. Photostimulated discharge in electret polymers: An alternative approach for investigating deep traps. *IEEE Trans. Dielectr. Electr. Insul.* **11**, 218-226 (2004).
 18. Mellinger, A., Gonzalez, F. C., Gerhard-Multhaupt, R., Santos, L. F., Faria, R. M. Photostimulated discharge of corona and electron-beam charged electret polymers. *IEEE ISDE Proceedings* (2002).
 19. Papaioannou, G., Coccetti, F., Plana, R. On the modeling of dielectric charging in RF-MEMS capacitive switches. *IEEE SiRF* 108-111 (2010).
 20. Turnhout, J. V. Thermally stimulated discharge of electrets. *Springer* (2005).
-

REVIEWER COMMENTS

Reviewer #2 (Remarks to the Author):

I am pleased to see that the authors have made a modest attempt at responding to the initial queries. However, there are still some serious confusions or problems.

1. I am somewhat confused with their data. As shown in Fig. 4c, the transmittance is ~96% (shall be somewhat higher) at the cathode current about 100 μA , while it is ~90% (actually a little lower) at its initial test under the same current as given in Fig. 4d. It is almost about 10% of discrepancy. Thus, I would like to know the repeatability of their experimental results.

2. I doubt of some data/calculation. Since the electron emission area is 0.08 cm^2 , how the emission current of 2.14 μA is corresponding to a current density of 0.037 mA/cm^2 ? In addition, compared with the data shown in previous version of Fig. 6, the calculated current density does not agree with the data in the current Fig. 6. The authors shall carefully check their data.

3. As for your first response, in the previous version of manuscript, the transmittance (91.84%) at 101.59 μA is relatively lower than that of revised version (~96%) at an even higher current (1 mA). Please clarify the reason.

4. Additionally, as shown in the test results of Fig. S10, the maximum current density of the modified cathode is 17.54 mA/cm^2 , compared to 1.27 mA/cm^2 before modification, representing an approximately 13-fold increase in current density. Is there any relationship between the increment of current density and transmittance?

5. Why do they define the turn-on field and threshold field corresponding to uncommon specific current of 0.8 μA and 80 μA , respectively? They shall follow general definition such as given in APL 106 (2015) 073501, Ceramic Intern. 47 (2021) 4034, etc. or the discussion is meaningless.

6. Why do the authors separate the transmittance vs current data into two subfigures (Fig. 4b and 4c)? Why not to show in one figure to save space?

7. What kind of technique or equipment do you utilize to measure the current under 1 second?

8. In figure 5, the authors shall choose a more suitable part as the inset to make more compatible with their legends.

Responses to Reviewer's Comments (NCOMMS-23-23925B)

Response to Reviewer 2

General comments:

I am pleased to see that the authors have made a modest attempt at responding to the initial queries. However, there are still some serious confusions or problems.

Response to General Comments:

We genuinely appreciate the time and effort you dedicated to reviewing our manuscript and for providing valuable feedback. Your recognition of our work is highly valued. To comprehensively address your comments and alleviate any concerns, we have made significant improvements to our manuscript. These improvements include the addition of new data, in-depth analyses, expanded discussions, and detailed clarifications (**with revised two Figures, added one Supplementary Figure, and seven Responded Figures**). In the following sections, we will respond to your comments point by point. Any newly added contents or changes in both the revised manuscript and Supplementary Information (SI) are clearly marked in **red and highlighted** for your convenience.

Specific Comments:

Comment 1:

I am somewhat confused with their data. As shown in Fig. 4c, the transmittance is ~96% (shall be somewhat higher) at the cathode current about 100 μ A, while it is ~90% (actually a little lower) at its initial test under the same current as given in Fig. 4d. It is almost about 10% of discrepancy. Thus, I would like to know the repeatability of their experimental results.

Response to comment 1:

We appreciate the reviewer's constructive comments and suggestions. In general, the data inconsistency is due to the differences between the modified cathode and the original prototype. In Fig. 4d, it is indeed observed that there is a decrease in the E-beam transmittance at the current about 100 μ A, which is the data of the first version cathode (V1). Thermal effects cause damage on the CNTs, seen as the changes of morphology. The damaged CNTs also deposit on the surface of the SiN_x-gate, resulting in a decrease of E-beam transmittance. In the latest cathode (V2), the interface structure between the CNTs emitter and the substrate has been modified to enable the cathode to work at higher current without significant thermal damages. Therefore, the E-beam transmittance can be maintained at ~96% for the latest cathode with the current up to 1 mA in Fig. 4c. The repeatability tests with three devices have also evidenced the data consistency. The details of analysis and experiment data are provided below.

- The discussion concerning the E-beam transmittance fluctuations in Fig. 4d of the first revision of the manuscript.

As shown in Fig. R1a, when the cathode (V1) is equipped with the CNTs emitter, we previously conducted tests on the emission current and E-beam transmittance. As the cathode emission current increases to $\sim 100 \mu\text{A}$, the E-beam transmittance of the cathode (V1) decreases from $\sim 95\%$ to $\sim 90\%$. Additionally, we subjected the cathode (V1) to a week-long endurance test at $\sim 100 \mu\text{A}$. The results in Fig. R1b (formerly Fig. 4d in the first revision) show that the cathode (V1) maintains an average emission current of $101.59 \mu\text{A}$ over the course of a week, with an average E-beam transmittance of 91.84% .

Fig. R1 The E-beam transmittance tests based on the cathode (V1). **a** The relationship between transmittance and cathode current of the cathode (V1). **b** Long-term current and transmittance measurements of the cathode (V1).

The presence of thermal effects can disrupt the emitter's normal operation^{1, 2}. As the voltage applied to the CNTs emitter increases, the CNTs emitter's current gradually rises, leading to a more pronounced manifestation of thermal effects at the edges of the CNTs emitter. Under the influence of thermal effects, some CNTs may fracture and evaporate. The fractured CNTs and evaporated carbon will also deposit on the surface of the SiN_x-gate due to the action of the electric field, increasing the likelihood of electrons interception by the SiN_x-gate, thereby reducing the E-beam transmittance of the cathode (V1). Furthermore, alterations in the morphology of the SiN_x-gate's surface can affect the distribution of the electric field on the SiN_x-gate's surface, consequently impacting the SiN_x-gate's E-beam focusing capabilities, which, in turn, affect the E-beam transmittance through the SiN_x-gate. When the cathode (V1) operates at around $100 \mu\text{A}$, it is operating beyond the working limit of the CNTs emitter, and the influence of thermal effects on the cathode (V1) becomes significant. Consequently, in Fig. R1b, the E-beam transmittance through the SiN_x-gate exhibits a certain reduction and fluctuation.

To further elucidate the underlying reasons for the E-beam transmittance instability observed in Fig. R1b, we conducted the characterizations of the SiN_x-gate's morphology after a week of continuous operation under $\sim 100 \mu\text{A}$. Fig. R2a-b depict the SEM morphologies of the SiN_x-gate before field emission (FE). It is evident from the images that the surface and sidewall morphologies of the SiN_x-gate are well-preserved. Fig. R2c-f show the SEM morphologies of the SiN_x-gate after one week of FE. It is discernible from the images that certain CNTs undergo fracture and evaporation due to the presence of thermal effects. These fractured and evaporated CNTs deposit on the surface and sidewall of the SiN_x-gate under the influence of the electric field.

Fig. R2 The cathode (V1) morphologies before and after long-term testing. **a** The top view and **(b)** side view of the SiN_x-gate before FE. **c** The top view and **(d)** side view of the SiN_x-gate after one week FE under ~100 μA. **e-f** The enlarged view of the SiN_x-gate after one week FE under ~100 μA. The carbon element content on the SiN_x-gate surface **(g)** before and **(h)** after one week FE under ~100 μA. **i** The morphology of CNTs emitter after one week FE under ~100 μA.

Furthermore, we conducted the elemental composition characterizations of the SiN_x-gate before and after FE. The carbon content on the SiN_x-gate surface is 2.24% before FE (Fig. R2g), while after FE, the carbon content on the SiN_x-gate surface increases to 44.05% (Fig. R2h). This further underscores the morphological changes on the SiN_x-gate surface. From the post-FE morphology of the CNTs emitter (Fig. R2i), it is apparent that some CNTs undergo damage due to thermal effects, which is a primary reason for the SiN_x-gate's morphological alterations.

In summary, the morphological degradation of some CNTs emitters in the cathode (V1) contributes to changes in the SiN_x-gate's appearance, thereby increasing the likelihood of electrons interception by the SiN_x-gate. These morphological changes on the SiN_x-gate can impact the distribution of the surface electric field and, consequently, the SiN_x-gate's electron-focusing efficiency. The characterizations and analyses presented above offer further insights into the factors contributing to the reduction and instability of E-beam transmittance through the SiN_x-gate.

- Improvement on the CNTs emitter for enhancing the cathode's performance at high emission currents.

In the last revision, the reviewers suggested that we should investigate the E-beam

transmittance of the cathode at high current/current density levels. Therefore, we made improvements to the second version cathode (V2) to enhance the cathode's current/current density range and their stability under high-current/current density conditions.

As shown in Fig. R3a, in the cathode (V1), the contact between the CNTs emitter and the metal substrate is physical in nature (the metal substrate encapsulates the CNTs emitter), and there is no chemical bonding between them. In this configuration, there is a certain air gap between the CNTs and the metal substrate. This results in the increases of the cathode's interface potential barrier and interface resistance, making the cathode susceptible to significant thermal effects when operating at high currents. In the cathode (V2), efforts have been made to reduce the resistance of the interface by modifying it into chemical bonding. We employed the Ti metal to modify the CNTs at the interface, at elevated temperatures, CNTs can react with Ti to form TiC. The generated TiC can establish chemical bonds between CNTs and the metal substrate (Fig. R3b), rather than relying on physical contact. This modification significantly reduces the cathode's interface potential barrier and interface resistance. Furthermore, previous research has indicated that, compared to pure CNTs emitter, modified CNTs@TiC emitter exhibits lower emission barrier and superior emission performances^{3,4}.

Fig. R3 The modification of the CNTs emitter. **a** The interface structure between CNTs emitter and substrate. **b** The interface structure between CNTs@TiC emitter and substrate. **c** The Raman spectra of CNTs emitter and CNTs@TiC emitter. **d** The interface resistances of CNTs emitter and CNTs@TiC emitter.

The CNTs emitter of the cathode (V1) and the CNTs@TiC emitter of the cathode (V2) were further examined by Raman spectra as shown in Fig. R3c. This analysis revealed the presence of two prominent peaks, namely the D band and the G band. The former is indicative of disordered carbon, encompassing defects and amorphous carbon, while the latter signifies the presence of covalent sp^2 bonds characteristic of graphite structures⁵. In Fig. R3c, the $I_{D'}/I_D$ ratio is merely 0.36, which suggests the reduction in

defects and impurity content of the CNTs@TiC emitter. This could be attributed to the reaction between certain amorphous carbon and Ti, resulting in the formation of TiC. Additionally, the elevated temperatures may lead to amorphous carbon evaporation within the CNTs emitter⁶. This experimental result also indicates that, in comparison to pure CNTs emitter, the modified CNTs@TiC emitter exhibits superior quality, and the probability of emitter damage due to defects is lower when operating at high currents.

Additionally, we conducted measurements of the interface resistances for both emitter types. The measurements indicate that the interface resistance for the unmodified CNTs emitter is 0.29 Ω , whereas the interface resistance for the modified CNTs emitter decreases to 0.06 Ω (Fig. R3d). Indeed, the generation of TiC transforms the physical contact between CNTs and the metal substrate into a chemical contact, consequently decreasing the interface resistance. The remarkably low interface resistance also diminishes the probability of emitter damage due to thermal effects when operating at high currents. All test results presented above collectively signify that the modified emitters can accommodate a broader range of cathode currents/current densities and exhibit superior stability under high-current/current density conditions. Consequently, when operating at high currents, the influence of the emitter on the SiN_x-gate is minimal, leading to reduced fluctuations in E-beam transmittance through the SiN_x-gate. The paper to discuss the modification of CNTs@TiC emitter is in preparation for another submission in the near future.

Based on the theoretical analyses and experimental results presented above, the modified CNTs@TiC emitter exhibits significantly enhanced material performances. Consequently, cathode (V2) based on CNTs@TiC emitter can operate stably at higher currents, with E-beam transmittance through the SiN_x-gate remaining relatively consistent, even as the current varies. As shown in Fig. R4 (formerly Fig. 4c in the first revision), we conducted current tests on the cathode (V2) with modified CNTs@TiC emitter. The results indicate that the emission current range for the cathode (V2) can reach from 2.14 μA to 1 mA (corresponding to the current density of 0.037 mA/cm²-17.54 mA/cm²), with an average E-beam transmittance of the cathode reaching ~96%.

Fig. R4 The relationship between transmittance and cathode current of the cathode (V2) based on the CNTs@TiC emitter.

To verify the long-term stability of the cathode (V2) based on the CNTs@TiC emitter under high current levels ($\sim 100 \mu\text{A}$), we conducted a prolonged stability test for the cathode (V2). As illustrated in the latest Fig. 4c, during a continuous 100 hours FE

test, the cathode (V2) consistently maintains an average cathode current of 99.40 μA , with a minimal current fluctuation of 2.5%. The mean anode current is 95.69 μA , with a minimal current fluctuation of 2.8%. Furthermore, the average E-beam transmittance through the SiN_x -gate is measured at 96.25%, exhibiting a minor transmittance fluctuation of 0.42%. Throughout the entire testing process, there is no noticeable degradation or fluctuation in the E-beam transmittance of the SiN_x -gate, indicating that the SiN_x -gate's electron-focusing effect remains constant.

Fig. 4c The 100 hours current stability of the cathode (V2) upon $\sim 100 \mu\text{A}$, the inset is an enlargement of the E-beam transmittance over a 20 s interval.

In addition, we conducted morphological characterizations of the SiN_x -gate and emitter after 100 hours emission for the cathode (V2). As depicted in Fig. R5a-c, the post-emission SiN_x -gate maintains favorable morphologies, with no apparent carbon adsorption or evaporation. Results in Fig. R5d-f also indicate that the CNTs@TiC emitter, even after extended emission, displays well-preserved morphologies without substantial damage. These characterizations outcome further reinforce the notion that the modified CNTs@TiC emitter can operate stably at high currents, with the impact on the E-beam transmittance of SiN_x -gate being negligible.

Fig. R5 The cathode (V2) morphologies after 100 hours testing. **a-c** The SiN_x -gate morphologies after 100 hours emission upon $\sim 100 \mu\text{A}$. **d-f** The CNTs@TiC emitter morphologies after 100 hours emission upon $\sim 100 \mu\text{A}$.

Hence, to prevent any confusion for readers, we have replaced Fig. 4d in the first

revision with the emission data at $\sim 100 \mu\text{A}$ for 100 hours (latest Fig. 4c) of the cathode (V2). Corresponding data have been corrected and marked in the latest manuscript.

Additionally, as shown in Fig. R4, when the emission current of the cathode (V2) reaches $\sim 1 \text{ mA}$, there is also a decreasing trend in the E-beam transmittance. This suggests that the improved CNTs@TiC emitter also has a certain emission current/current density limit. When the emission current of these emitters exceeds the emission limit, the instability of the emitters can affect the normal operation of the cathode and the SiN_x-gate.

Therefore, to further enhance the cathode's emission current range to a few mA, it is advisable to increase the CNTs@TiC emitter's area rather than increasing the CNTs@TiC emitter's current density. As shown in Supplementary Fig. 18, to further increase the cathode's emission current, we connected multiple cathodes in parallel operation, resulting in a cathode emitter area of 0.342 cm^2 . The total emission current of the parallel-connected cathode can reach $\sim 3.25 \text{ mA}$ (corresponding to the current density of 9.50 mA/cm^2), and the average E-beam transmittance of the cathode remains above 95% during 60 hours of continuous operation.

Supplementary Fig. 18 Long-term current measurements of the cathode (V2) with SiN_x-gate. **a** Emission current stability of the cathode at 3.25 mA, the inset illustrates the current fluctuations of I_c and I_a within 2 minutes. **b** Transmittance of the cathode, the inset is the E-beam transmittance fluctuation within 2 minutes. The emitter area is 0.342 cm^2 .

- The repeatability tests of the cathodes (V2) based on the modified CNTs emitter.

To further validate the reliability and repeatability of our data, we prepared three cathode (V2) samples based on the CNTs@TiC emitter, denoted as sample #1, sample #2, and sample #3, respectively. We individually measured the I - E curves for these three

samples in the 0-100 μA range and calculated the corresponding E-beam transmittances at different currents (Fig. R6a-f). As depicted in these figures, the average transmittances (T_{avg}) in the 0-100 μA range for sample #1, sample #2, and sample #3 stand at 96.32%, 96.01%, and 96.05%, respectively. These results exhibit good repeatability.

Fig. R6 The repeatability tests of the E-beam transmittance based on the modified CNTs@TiC emitter. **a** The I - E curve and **(b)** the transmittance versus cathode current plot of sample #1. **c** The I - E curve and **(d)** the transmittance versus cathode current plot of sample #2. **e** The I - E curve and **(f)** the transmittance versus cathode current plot of sample #3.

To verify the repeatability of the cathodes (V2) based on the CNTs@TiC emitter at currents of around 100 μA , we conducted 24 hours current tests on the three aforementioned samples. Under consistent testing conditions, we measured the emission currents for these three samples and calculated the E-beam transmittances variation over the course of 24 hours. Fig. R7a-f present the current and transmittance data obtained using different cathode (V2) samples. It is evident from these figures that, despite some fluctuations in emission current, the E-beam transmittances through the SiN_x-gate remain relatively stable. This can be attributed to the stable operation of the modified CNTs@TiC emitter. Furthermore, the T_{avg} within 24 hours for sample #1, sample #2, and sample #3 are 95.33%, 96.30%, and 95.96%, respectively, corresponding well with the E-beam transmittance data obtained for the samples at around 100 μA in Fig. R6. This further underscores the repeatability and stability of the E-beam transmittance through the SiN_x-gate.

Fig. R7 The repeatability tests of the E-beam transmittance based on the modified CNTs@TiC emitter. **a** The emission currents and **(b)** corresponding transmittance within 24 hours of sample #1. **c** The emission currents and **(d)** corresponding transmittance within 24 hours of sample #2. **e** The emission currents and **(f)** corresponding transmittance within 24 hours of sample #3.

● Changes in the manuscript.

[On Page 14 and Page 16 of the revised manuscript (Fig. 4c)]

Moreover, to verify the long-term stability of the cathode based on the SiN_x-gate under high current levels (~100 µA), we conducted a prolonged stability test for cathode emission. As illustrated in Fig. 4c, during a continuous 100 hours FE test, the cathode consistently maintains an average I_c of 99.40 µA, with a minimal current fluctuation of 2.5%. The mean I_a is 95.69 µA, with a minimal current fluctuation of 2.8%. Furthermore, the average E-beam transmittance through the SiN_x-gate is measured at 96.25%, exhibiting a minor transmittance fluctuation of 0.42%. Throughout the entire testing process, there is no noticeable degradation or fluctuation in the E-beam transmittance of the SiN_x-gate, indicating that the SiN_x-gate's electron-focusing effect remains constant.

Fig. 4c The 100 hours current stability of the cathode upon $\sim 100 \mu\text{A}$, the inset is an enlargement of the E-beam transmittance over a 20 s interval.

Comment 2:

I doubt of some data/calculation. Since the electron emission area is 0.08 cm^2 , how the emission current of $2.14 \mu\text{A}$ is corresponding to a current density of 0.037 mA/cm^2 ? In addition, compared with the data shown in previous version of Fig. 6, the calculated current density does not agree with the data in the current Fig. 6. The authors shall carefully check their data.

Response to comment 2:

We greatly appreciate the reviewer for pointing out the mistake. We apologize for any inappropriate description regarding the emitter area and have made the necessary revisions and additions to provide accurate information about the emitter area in the revised manuscript. Additionally, we have provided explanations for the differences in the maximum current density values in different versions of Fig. 6. Finally, we have conducted a comprehensive review and verification of all the data presented in the revised manuscript and SI to ensure data accuracy.

The emitter area of 0.08 cm^2 corresponds to the area of the CNTs emitter in cathode (V1), whereas the modified CNTs emitter in cathode (V2) has a designed area of 0.057 cm^2 . The cathode current data from $2.14 \mu\text{A}$ (the actual measurement value is $2.136 \mu\text{A}$, and $2.14 \mu\text{A}$ is the result after rounding to two decimal places) to 1 mA (Fig. R4) were measured using the cathode (V2) based on the modified CNTs emitter, resulting in the corresponding current density values of 0.037 mA/cm^2 to 17.54 mA/cm^2 . To provide a clearer representation of our experimental parameters, we have made supplementary and improved notations regarding the emitter area used in our latest manuscript and the SI.

In the initial manuscript, we did not modify the CNTs emitter. At that time, the emission current range of the cathode (V1) is around $0\text{-}101.59 \mu\text{A}$ (Fig. R1a), and the emitter area is 0.08 cm^2 , resulting in the corresponding current density of $0\text{-}1.27 \text{ mA/cm}^2$. In the first revision of the manuscript, we modified the emitter to increase the current range. After modification, the emission current range of the cathode (V2) is $2.14 \mu\text{A}\text{-}1 \text{ mA}$ (Fig. R4), and the designed emitter area is 0.057 cm^2 , leading to the corresponding current density of $0.037 \text{ mA/cm}^2\text{-}17.54 \text{ mA/cm}^2$. Therefore, in the first revision of the manuscript, we updated the maximum current density value in Fig. 6 to

17.54 mA/cm².

Fig. 6 Comparison of performances of the proposed cathode among various gate-type cathodes based on CNTs emitter.

- Changes in the manuscript and SI.

[On Page 14 Line 11 of the revised manuscript]
The emitter area is 0.057 cm².

[On Page 17 Line 12 of the revised manuscript]
The emitter area is 0.08 cm².

[On Page 27 Line 5 of the revised SI]
The emitter area is 0.342 cm².

[On Page 39 Line 14 of the revised SI]
The emitter area is 0.057 cm².

Comment 3:

As for your first response, in the previous version of manuscript, the transmittance (91.84%) at 101.59 μ A is relatively lower than that of revised version (~96%) at an even higher current (1 mA). Please clarify the reason.

Response to comment 3:

We thank the reviewer for the constructive comments. As mentioned above, the changes in E-beam transmittances between the previous version and the revised version can be attributed to the differences in the CNTs emitters. The reduction in average E-beam transmittance to 91.84% in the last version (Fig. R1b) can be attributed to the instability of the CNTs emitter in the cathode (V1). For the CNTs emitter of the cathode (V1), the emission current limit is around 100 μ A (corresponding to the current density of

approximately 1.25 mA/cm²). When the emission current exceeds the CNTs emitter's emission limit, thermal effects cause damage to the CNTs emitter's morphology, subsequently affecting the SiN_x-gate's morphology and E-beam transmittance. To further enhance the stability of the CNTs emitter and increase the current/current density range, we modified the CNTs emitter in the cathode (V2). The modified emitter of the cathode (V2) has an emission current range of 2.14 μA-1 mA (corresponding to the current density of 0.037 mA/cm²-17.54 mA/cm²), and the average E-beam transmittance of approximately 96% (Fig. R4). To avoid confusion for readers, we have conducted 100 hours testing using the cathode (V2) with modified CNTs emitter at approximately 100 μA (latest Fig. 4c) and replaced the average E-beam transmittance result of 91.84% of the cathode (V1) in the previous version of manuscript (Fig. R1b).

Comment 4:

Additionally, as shown in the test results of Fig. S10, the maximum current density of the modified cathode is 17.54 mA/cm², compared to 1.27 mA/cm² before modification, representing an approximately 13-fold increase in current density. Is there any relationship between the increment of current density and transmittance?

Response to comment 4:

We thank the reviewer for the constructive comments. As mentioned above, the increase in cathode current density can be attributed to our modification of the CNTs emitter. Compared to the unmodified CNTs emitter in the cathode (V1), the modified CNTs emitter in the cathode (V2) can operate stably at higher current densities and does not adversely affect the morphology and E-beam transmittance of the SiN_x-gate.

Moreover, due to the limited trap density of electret materials, after SiN_x electret reaches charging saturation, the surface potential of the SiN_x-gate will cease to change with increasing charging time⁷. Similarly, upon achieving charging saturation in the SiN_x-gate, even if we increase the negative high voltage on the cathode to increase the cathode emission current/current density, theoretically, the surface potential of the SiN_x-gate will not change. This is because, at this stage, the electrons trapped within the SiN_x have reached saturation, leading to a stable surface potential (Supplementary Fig. 13a). To further validate this, we first charged the SiN_x electret material. Once saturation was achieved, we gradually increased the charging voltage and measured the resulting changes in surface potential. Experimental results indicate that after charging saturation, the impact of voltage on the surface potential of SiN_x is minimal (Supplementary Fig. 13b).

Supplementary Fig. 13 The charging saturation of SiN_x electret. **a** The illustration for charging saturation of SiN_x electret. **b** Variation of the normalized surface potential of SiN_x electret with charging voltage curve after the charging saturation of the SiN_x electret.

Based on the above analysis, it can be observed that after saturation of the charging of the SiN_x-gate, increasing the cathode voltage does not result in any significant change in the surface potential of the SiN_x-gate. Building upon this foundation, while maintaining the surface potential of the SiN_x-gate constant, we gradually increased the cathode emitter potential to investigate changes in the trajectories of emitted electrons from the cathode. As depicted in Supplementary Fig. 14, it can be observed that with the SiN_x-gate potential unchanged, as the cathode emitter potential increases from -500 V to -700 V, the SiN_x-gate is capable of achieving E-beam focusing, thereby achieving a high E-beam transmittance. Consequently, following saturation of charging in the SiN_x-gate, increasing the cathode emission current/current density by enhancing the cathode emitter potential will not result in significant alterations to the E-beam trajectories from the cathode.

Supplementary Fig. 14 Electrons trajectories at different cathode potentials of (a) -500 V, (b) -600 V, and (c) -700 V when SiN_x-gate potential remains constant.

Following the saturation of charging in SiN_x-gate, we conducted measurements of E-beam transmittance through the SiN_x-gate for cathode emission current ranging from 2.14 μ A to 1 mA (corresponding to the emission current density of 0.037 mA/cm² to 17.54 mA/cm²). As depicted in Fig. R4, the test results indicate that during the process of increasing the emission current/current density by enhancing the cathode voltage, the E-beam transmittance through the SiN_x-gate remains insignificantly altered. The average E-beam transmittance across the SiN_x-gate from 2.14 μ A to 1 mA is measured above 96%.

Both the conducted tests and simulations affirm that throughout the process of increasing the emission current density by enhancing the cathode emitter potential, there is no substantial change in the surface potential of the SiN_x-gate, consequently preserving the high and stable E-beam transmittance through the SiN_x-gate. It's essential to note that when the cathode emission current density exceeds the CNTs

emitter's emission limit, thermal effects can impact the CNTs emitter's stability. This instability of the CNTs emitter, in turn, affects the SiN_x-gate's E-beam transmittance. Therefore, increasing the CNTs emitter's current density won't impact the SiN_x-gate's high electron transmittance as long as it remains below the CNTs emitter's current density limit. When the CNTs emitter current density surpasses the emission limit, fluctuations in the SiN_x-gate's E-beam transmittance can occur. To achieve a greater cathode emission current range, it should be accomplished by increasing the CNTs emitter's area rather than continuously raising the CNTs emitter's current density.

Comment 5:

Why do they define the turn-on field and threshold field corresponding to uncommon specific current of 0.8 μA and 80 μA, respectively? They shall follow general definition such as given in APL 106 (2015) 073501, Ceramic Intern. 47 (2021) 4034, etc. or the discussion is meaningless.

Response to comment 5:

We thank the reviewer for the constructive comments and suggestions. In response to your comments and suggestions, we have redefined the cathode's turn-on field (E_{to}) and threshold field (E_{th}) following the general definition in prior research findings. E_{to} is defined as the electric field required to achieve a cathode current density of 10 μA/cm², and E_{th} is defined as the required electric field corresponding to a cathode current density of 1 mA/cm².

- Changes in the manuscript.

[On Page 15 of the revised manuscript]

The turn-on electric field E_{to} (corresponding to cathode current density of 10 μA/cm²) and threshold electric field E_{th} (corresponding to cathode current density of 1 mA/cm²) can be obtained from the I - E curve and are found to be about 0.41 V/μm and 0.61 V/μm, respectively^{8,9}.

Comment 6:

Why do the authors separate the transmittance vs current data into two subfigures (Fig. 4b and 4c)? Why not to show in one figure to save space?

Response to comment 6:

We thank the reviewer for the suggestions. In response to your suggestions, we have consolidated the previous Fig. 4b and Fig. 4c into a latest Fig. 4b to save space. As depicted in the latest Fig. 4b, we have re-measured the I - E curve using the cathode (V2) with modified CNTs emitter. The inset in Fig. 4b provides a comparison of E-beam transmittances between cathodes (V2) based on SiN_x-gate and Au-gate, the current data were also re-measured using the modified CNTs emitter.

- Changes in the manuscript.

[On Pages 14-16 of the revised manuscript (Fig. 4b)]

As shown in Fig. 4b, the average E-beam transmittance remains high at 96.05% within about 0-100 μA range (corresponding to a current density of 0-1.75 mA/cm^2) according to the FE curves, indicating that the SiN_x -based gate transmittance is robust under varying emission currents. To validate the E-beam transmittance of the cathode at higher current levels, following the saturation of charging in SiN_x -gate, we conducted measurements of E-beam transmittance through the SiN_x -gate for cathode emission current ranging from 2.14 μA to 1 mA (corresponding to the emission current density of 0.037 mA/cm^2 to 17.54 mA/cm^2). As depicted in the inset of Fig. 4b, the test results indicate that during the process of increasing the emission current by enhancing the cathode voltage, the E-beam transmittance through the SiN_x -gate remains insignificantly altered. The average E-beam transmittance across the SiN_x -gate from 2.14 μA to 1 mA is measured at 96.33%. In contrast, when the FE performances of the Au-gate were evaluated at various emission currents using the same experimental setup, the average E-beam transmittance drops to 27.21% within the 2.50 μA -1 mA range due to the Au-gate interception and space-charge effect during the FE process depicted in the inset of Fig. 4b¹⁰. The average E-beam transmittance of the SiN_x -based cathode is increased by 254% to about 96.33% at about 0-1 mA current range owing to the electret material's electrostatic focusing effect.

Fig. 4b The I - E curve of the cathode based on the modified CNTs emitter, the inset is a comparison of E-beam transmittances between cathodes based on SiN_x -gate and Au-gate.

Comment 7:

What kind of technique or equipment do you utilize to measure the current under 1 second?

Response to comment 7:

We thank the reviewer for the comments. To provide a more comprehensive explanation of the measurement process for current data, we have added detailed descriptions of the equipment and devices utilized during the measurement procedures. It's important to

note that during the current measurement process for studying the charging time of the SiN_x-gate, the sampling rate of the current signals is set to 2000 Hz, which allows for clear measurement of changes in the cathode's emission current within a 1 s interval.

- Changes in the manuscript and SI.

[On Page 13 of the revised manuscript and Page 19 of the revised SI (Supplementary Fig. 11)]

Fig. 11a illustrates the schematic diagram of the cathode emission current measurement system. As depicted in the diagram, when a negative high-voltage source (Wisman, DL5N300) is applied as the voltage input to the cathode emitter and the SiN_x-gate, the current generated by the emitter is successively intercepted and collected by the SiN_x-gate and the anode. The current signals collected by the SiN_x-gate and anode are initially output through the high-voltage flange of the vacuum system to the electrometer (Keithley, 6514). The electrometer can accommodate current signals ranging from 0 to 20 mA. Subsequently, the electrometer transmits the collected current signals to the data acquisition (DAQ) system (including the NI BNC-2121 Connector and the NI PCI-6250 Multifunction Acquisition Module). The DAQ system is capable of converting analog current signals into digital signals for reading and storage, with a sampling rate that can reach up to ~kHz. Ultimately, we can monitor the collected current data on the computer. Fig. 11b provides a visual representation of the cathode emission current measurement system.

Supplementary Fig. 11 The current measurement system of the cathode. **a** The schematic diagram of the cathode emission current measurement system. **b** The image of the cathode emission current measurement system.

Comment 8:

In figure 5, the authors shall choose a more suitable part as the inset to make more compatible with their legends.

Response to comment 8:

We thank the reviewer for the constructive comments and suggestions. We have selected a more appropriate section as the inset to ensure better compatibility with the

legends in Fig. 5. From the inset in Fig. 5a, it can be observed that while there are slight fluctuations in the cathode current and anode current at approximately 26.5 μA and 25.5 μA , respectively, the emission remains generally stable over the course of 550 hours of operation. From the inset in Fig. 5b, it's evident that although there is a minor fluctuation in the SiN_x-gate's E-beam transmittance, the E-beam transmittance remains relatively stable throughout the 550 hours of emission and can be maintained at around 96.17%.

- Changes in the manuscript.

[On Pages 17-18 of the revised manuscript (Fig. 5)]

The insert in Fig. 5a shows an enlarged view of I_c and I_a within 1 minute. The insert in Fig. 5b provides an enlarged view of E-beam transmittance within 1 minute.

Fig. 5 Long-term current measurements of the cathode with SiN_x-gate. **a** Emission current stability of the cathode, the inset illustrates the current fluctuations of I_c and I_a within 1 minute. **b** Transmittance of the cathode, the inset is the E-beam transmittance fluctuation within 1 minute. **c** Surface morphologies of the CNTs emitter before and after long-term testing. **d** Surface morphologies of the SiN_x-gate before and after long-term testing. The emitter area is 0.08 cm².

We sincerely thank you for your insightful comments and valuable suggestions again, which have considerably improved the quality of our manuscript. We believe that our revised manuscript, in conjunction with our responses, effectively addresses your concerns and comments. We appreciate your time and consideration.

References

1. Williams, L. T., Kumsomboone, V. S., Ready, W. J., Walker, M. L. R. Lifetime and failure mechanisms of an arrayed carbon nanotube field emission cathode. *IEEE Trans. Electron Devices* **57**, 3163-3168 (2010).
2. Dean, K. A., Burgin, T. P., Chalamala, B. R. Evaporation of carbon nanotubes during electron field emission. *Appl. Phys. Lett.* **79**, 1873-1875 (2001).
3. Qin, Y., Hu, M. Characterization and field emission characteristics of carbon nanotubes modified by titanium carbide. *Appl. Surf. Sci.* **254**, 3313-3317 (2008).
4. Zang, J. B., Lu, J., Wang, Y. H., Zhang, J. H., Cheng, X. Z., Huang, H. Fabrication of core-shell structured MWCNT-Ti (TiC) using a one-pot reaction from a mixture of TiCl_3 , TiH_2 , and MWCNTs. *Carbon* **48**, 3802-3806 (2010).
5. Tsai, W. L., et al. Conductivity enhancement of multiwalled carbon nanotube thin film via thermal compression method. *Nanoscale Res. Lett.* **9**, 1-6 (2014).
6. Wang, M., et al. Wafer-scale transfer of vertically aligned carbon nanotube arrays. *JACS* **136**, 18156-18162 (2014).
7. Leonov, V., Hoof, C. V., Goedbloed, M., Schaijk, R. V. Charge injection and storage in single-layer and multilayer inorganic electrets based on SiO_2 and Si_3N_4 . *IEEE Trans. Dielectr. Electr. Insul.* **19**, 1253-1260 (2012).
8. Xu, J., et al. Outstanding field emission properties of wet-processed titanium dioxide coated carbon nanotube based field emission devices. *Appl. Phys. Lett.* **106**, 073501 (2015).
9. Wu, D., et al. Field emission from geometrically modulated tungsten-nickel sulfide/graphitic carbon nanobelts on Si microchannel plates. *Ceram. Int.* **47**, 4034-4042 (2021).
10. Hruby, V., Gasdaska, C., Falkos, P., Delichatsios, M., Rostler, P. Modeling of field emission cathodes for space propulsion. *AIAA 33rd Plasmadynamics Lasers Conf.* 2124 (2002).

REVIEWERS' COMMENTS

Reviewer #2 (Remarks to the Author):

The authors have well addressed all of my concerns and better performance devices are presented in the revised manuscript. Thus, I would like to support it for publication in this prestigious journal.

Responses to Reviewer's Comments (NCOMMS-23-23925C)

Response to Reviewer 3

Remarks:

The effect of dielectric material properties on charging, which are determined by fabrication.

Response:

We thank the reviewer for the constructive comments. The charging performances of dielectric materials can be influenced certainly by the fabrication process. As reported by U. Zaghoul et al., gas ratio, RF power, and substrate temperature have been found to influence the trap distributions in SiN_x , thereby affecting the charging and discharging processes¹. The reactive gas ratio has been observed to have a more significant impact on the stoichiometry of SiN_x material compared to the effects of RF power or substrate temperature from the FT-IR results shown in Fig. R1a-c. As the SiH_4/NH_3 gas ratio increases from 0.15 to 0.45, the injected charge density and relaxation time constant gradually decrease. However, for gas ratios of 0.6 and 0.8, the decrease becomes more pronounced. This trend can be attributed to the increased silicon content in the SiN_x films, as confirmed by FT-IR data. The higher silicon content results in a higher leakage current, a greater concentration of defects, and an increased number of charge-trapping centers. Consequently, the charge redistribution paths become more extensive with higher silicon content in the SiN_x film, leading to a larger redistribution current and a smaller charge density at the dielectric surface¹.

Fig. R1 FT-IR data comparison for SiN_x films deposited using different (a) gas ratios, (b) RF powers, and (c) substrate temperatures¹.

Moreover, the resulting potential profile for the SiN_x samples deposited using different reactive gas ratios is shown in Fig. R2, where charges were injected under the same pulse amplitude (40 V)¹. It can be observed from the figure that surface potential (U_s), full-width at half maximum (FWHM), and consequently, the integral of the potential profile, which directly indicates the injected charge density inside the dielectric films exhibit significant variations with respect to the deposition gas ratio r .

Fig. R2 The extracted potential profiles for SiN_x samples under different gas ratios r^1 .

In summary, the variations in gas ratio, RF power, and substrate temperature can impact the trap level distribution in SiN_x films. As shown in Supplementary Fig. 7, with higher defect concentrations, there are more charge-trapping centers, resulting in multiple charge redistribution paths within the SiN_x material. Hence, the charging performances of dielectric materials can be certainly influenced by the fabrication process. The effect of SiN_x electret deposition parameters on material properties was also investigated in our study, and the optimal deposition parameters were determined based on FT-IR characterization results. Finally, we fabricated the SiN_x electret using the optimized deposition parameters.

Supplementary Fig. 7 The trap level distributions of SiN_x with different charge-trapping centers.

The relationship between the charging time of electret materials and their surface potential has been investigated by V. Leonov et al., which demonstrates that due to the limited trap density of electret materials, after SiN_x electret reaches charging saturation, the surface potential of the electret materials will cease to change with increasing charging time². As depicted in Fig. R3a, an increase in charging time leads to a gradual rise in the surface potential of various electret materials until reaching saturation. The Si₃N₄ electret in Fig. R3b exhibits a maximum surface potential of approximately -65 V after charging for 40 s, and the surface potential of Si₃N₄ electret remains relatively stable as the charging time increases, indicating that it has reached a state of charging saturation. Based on this analysis, we measured that the maximum surface potential of SiN_x-gate after charging saturation is approximately -60 V (Fig. R3b), corresponding

well with the measurement result by V. Leonov et al.

Fig. R3 (a) The relationship between the charging time of electret materials and their surface potential². (b) The SiN_x electret surface potential decay versus time.

Similarly, upon achieving charging saturation in the SiN_x-gate, theoretically, the surface potential of the SiN_x-gate will not change even if we increase the negative high voltage on the cathode to increase the cathode emission current/current density. This is because, at this stage, the electrons trapped within the SiN_x have reached saturation, leading to a stable surface potential (Supplementary Fig. 13a). To further validate this, we firstly charged the SiN_x electret material. Once saturation was achieved, we gradually increased the charging voltage and measured the resulting changes in surface potential. Experimental results indicate that after charging saturation, the impact of voltage on the surface potential of SiN_x is minimal (Supplementary Fig. 13b).

Supplementary Fig. 13 The charging saturation of SiN_x electret. **a** The illustration for charging saturation of SiN_x electret. **b** Variation of the normalized surface potential of SiN_x electret with charging voltage curve after the charging saturation of the SiN_x electret.

Based on the above analysis, it can be observed that after saturation of the charging of the SiN_x-gate, increasing the cathode voltage does not result in any significant change in the surface potential of the SiN_x-gate. Building upon this foundation, while maintaining the surface potential of the SiN_x-gate constant, we gradually increased the cathode emitter potential to investigate changes in the trajectories of emitted electrons from the cathode. As depicted in Supplementary Fig. 14a-c, it can be observed that with the SiN_x-gate potential unchanged, as the cathode emitter potential increases from -500 V to -700 V, the SiN_x-gate is capable of achieving E-beam focusing, thereby achieving a high E-beam transmittance. Consequently, following saturation of charging in the

SiN_x-gate, increasing the cathode emission current/current density by enhancing the cathode emitter potential will not result in significant alterations to the E-beam trajectories from the cathode.

Supplementary Fig. 14 Electrons trajectories at different cathode potentials of (a) - 500 V, (b) -600 V, and (c) -700 V when SiN_x-gate potential remains constant.

Following the saturation of charging in SiN_x-gate, we conducted measurements of E-beam transmittance through the SiN_x-gate under the cathode emission current ranging from 2.14 μA to 1 mA (corresponding to the emission current density of 0.037 mA cm^{-2} to 17.54 mA cm^{-2}). The test results indicate that during the process of increasing the emission current/current density by enhancing the cathode voltage, the E-beam transmittance through the SiN_x-gate remains insignificantly altered. The average E-beam transmittance across the SiN_x-gate from 2.14 μA to 1 mA is measured above 96%.

In summary, the charging performances of electret materials can be influenced by the fabrication process. However, once the fabrication process of electret material is determined, its surface potential remains relatively unchanged after reaching charging saturation. Moreover, these increases in charging time and charging voltage have minimal impact on its surface potential. Therefore, when increasing the cathode current/current density after charging saturation by increasing the cathode voltage, it is possible to maintain a stable surface potential for the SiN_x-gate. Based on this premise, the SiN_x-gate can effectively focus electrons and ensure high electrons transmittance among a wide range of currents/current densities.

- Changes in the manuscript.

[On Page 16 of the revised SI]

In summary, the variations in gas ratio, RF power, and substrate temperature can impact the trap level distributions in SiN_x film, thereby effecting the charging performances of the gate based on the SiN_x film.

[On Page 24 of the revised SI]

Though the charging performances of dielectric materials can be influenced by the fabrication process, its surface potential remains relatively unchanged after reaching

charging saturation once the fabrication process of electret material is determined. Moreover, these increases in charging time and charging voltage have minimal impact on its surface potential. Therefore, when increasing the cathode current/current density after charging saturation by increasing the cathode voltage, it is possible to maintain a stable surface potential for the SiN_x-gate. Based on this premise, the SiN_x-gate can effectively focus electrons and ensure high electrons transmittance among a wide range of currents/current densities.

We sincerely thank you for your insightful remarks and valuable suggestions again, which have considerably improved the quality of our manuscript. We believe that our revised manuscript, in conjunction with our responses, effectively addresses your concerns. We appreciate your time and consideration.

References

1. Zaghloul, U., et al. Effect of deposition gas ratio, RF power, and substrate temperature on the charging/discharging processes in PECVD silicon nitride films for electrostatic NEMS/MEMS reliability using atomic force microscopy. *JMEMS* **20**, 1395-1418 (2011).
2. Leonov, V., Hoof, C. V., Goedbloed, M., Schaijk, R. V. Charge injection and storage in single-layer and multilayer inorganic electrets based on SiO₂ and Si₃N₄. *IEEE Trans. Dielectr. Electr. Insul.* **19**, 1253-1260 (2012).